# Pooled-matrix protein interaction screens using Barcode Fusion Genetics

Nozomu Yachie[1,2,3,4,5,*,†], Evangelia Petsalaki[1,2,†], Joseph C Mellor[1,2], Jochen Weile[1,2,6], Yves Jacob[7], Marta Verby[1,2], Sedide B Ozturk[1,2], Siyang Li[1,2], Atina G Cote[1,2], Roberto Mosca[8], Jennifer J Knapp[1,2], Minjeong Ko[1,2], Analyn Yu[1,2], Marinella Gebbia[1,2], Nidhi Sahni[9,10,‡], Song Yi[9,10], Tanya Tyagi[1,2], Dayag Sheykhkarimli[1,2,6], Jonathan F Roth[1,2,6], Cassandra Wong[1,2], Louai Musa[1,2], Jamie Snider[1], Yi-Chun Liu[1], Haiyuan Yu[11], Pascal Braun[9,10,12], Igor Stagljar[1,6], Tong Hao[9,10], Michael A Calderwood[9,10], Laurence Pelletier[2,6], Patrick Aloy[8,13], David E Hill[9,10], Marc Vidal[9,10] & Frederick P Roth[1,2,6,9,14,15,**]

## Abstract

High-throughput binary protein interaction mapping is continuing to extend our understanding of cellular function and disease mechanisms. However, we remain one or two orders of magnitude away from a complete interaction map for humans and other major model organisms. Completion will require screening at substantially larger scales with many complementary assays, requiring further efficiency gains in proteome-scale interaction mapping. Here, we report Barcode Fusion Genetics-Yeast Two-Hybrid (BFG-Y2H), by which a full matrix of protein pairs can be screened in a single multiplexed strain pool. BFG-Y2H uses Cre recombination to fuse DNA barcodes from distinct plasmids, generating chimeric protein-pair barcodes that can be quantified via next-generation sequencing. We applied BFG-Y2H to four different matrices ranging in scale from ~25 K to 2.5 M protein pairs. The results show that BFG-Y2H increases the efficiency of protein matrix screening, with quality that is on par with state-of-the-art Y2H methods.

**Keywords** DNA barcode; interactome; next-generation sequencing; protein interaction; yeast two-hybrid

**Subject Categories** Methods & Resources; Network Biology
**Mol Syst Biol.** (2016) 12: 863

## Introduction

The impact of individual genotypes on disease is largely mediated by interactions between proteins. Protein interaction network mapping has shed light on cellular processes and disease mechanisms (Vidal *et al*, 2011). Some technologies (Rigaut *et al*, 1999; Gillet *et al*, 2012; Roux *et al*, 2013) yield indirect associations between proteins, while yeast two-hybrid (Y2H; Fields & Song, 1989) and other technologies (Kerppola, 2006; Tarassov *et al*, 2008; Cassonnet *et al*, 2011) tend to reveal direct physical ("binary") interactions (Rolland *et al*, 2014). Systematic high-quality Y2H has now revealed more protein interactions than the combination of all similar-quality interactions curated from smaller-scale studies in the scientific literature (Rolland *et al*, 2014). However, we remain an order of magnitude away from a complete map of human protein

1 Donnelly Centre, University of Toronto, Toronto, ON, Canada
2 Lunenfeld-Tanenbaum Research Institute, Mt. Sinai Hospital, Toronto, ON, Canada
3 Synthetic Biology Division, Research Center for Advanced Science and Technology, The University of Tokyo, Tokyo, Japan
4 Institute for Advanced Bioscience, Keio University, Tsuruoka, Yamagata, Japan
5 PRESTO, Japan Science and Technology Agency (JST), Tokyo, Japan
6 Department of Molecular Genetics, University of Toronto, Toronto, Ontario, Canada
7 Département de Virologie, Unité de Génétique Moléculaire des Virus à ARN, Institut Pasteur, Paris, France
8 Joint IRB-BSC Program in Computational Biology, Institute for Research in Biomedicine (IRB Barcelona), Barcelona, Spain
9 Center for Cancer Systems Biology (CCSB) and Department of Cancer Biology, Dana-Farber Cancer Institute, Boston, MA, USA
10 Department of Genetics, Harvard Medical School, Boston, MA, USA
11 Weill Institute for Cell and Molecular Biology, Cornell University, Ithaca, NY, USA
12 Department of Plant Systems Biology, Technische Universität München, Wissenschaftszentrum Weihenstephan, Freising, Germany
13 Institució Catalana de Recerca i Estudis Avançats (ICREA), Barcelona, Spain
14 Canadian Institute for Advanced Research, Toronto, ON, Canada
15 Department of Computer Science, University of Toronto, Toronto, Ontario, Canada
*Corresponding author. Tel: +81 3 5452 5242; E-mail: yachie@synbiol.rcast.u-tokyo.ac.jp
**Corresponding author. Tel: +1 416 946 5130; E-mail: fritz.roth@utoronto.ca
†These authors contributed equally to this work
‡Present address: Department of Systems Biology, The University of Texas MD Anderson Cancer Center, Houston, TX, USA

interactions (Rolland *et al*, 2014), and further still when we consider that different "proteoforms" (Smith & Kelleher, 2013) can interact with distinct partners (Corominas *et al*, 2014). Interaction maps are even less complete for most model organisms.

In Y2H, the transcription factor Gal4 is reconstituted via interaction of a "bait" protein fused to the Gal4 DNA-binding domain with a "prey" protein fused to the Gal4 activation domain (Fields & Song, 1989). For each tested pair of proteins, a strain is generated, encoding a specific combination of bait and prey. Interactions are then detected via selection for expression of a Gal4-activated reporter gene (Vidal & Legrain, 1999; Legrain & Selig, 2000; Uetz, 2002). A single bait strain can be mated to a pool of hundreds of prey strains (Rual *et al*, 2005; Yu *et al*, 2008; Simonis *et al*, 2009; Venkatesan *et al*, 2009), simplifying primary screening, but requiring later identification of Y2H-positive colonies. Costs of identification have been reduced by the Stitch-seq method (Yu *et al*, 2011); however, Stitch-seq is labor intensive, requiring isolation of individual Y2H-positive colonies and three individual PCRs for each colony. Another "Y2H-seq" (Weimann *et al*, 2013) approach identifies prey interactors from a pool by deep sequencing, but further retesting is required to identify the specific bait protein which interacts with each identified prey.

An ideal multiplexing strategy would allow efficient identification of each bait–prey combination within a pool of strains that collectively represents the entire protein-pair matrix. To this end, two combinatorial multiplexing strategies have been proposed in which the open reading frames (ORFs) for bait and prey proteins are physically linked: (i) intracellular DNA recombination (Hastie & Pruitt, 2007) and (ii) overlap-extension PCR within emulsion-encapsulated single cells (Nirantar & Ghadessy, 2011). However, these methods have never been implemented at large scale. Moreover, these methods require multiplexed PCR amplification of templates varying widely in length and base composition, a procedure subject to severe PCR competition effects (Shiroguchi *et al*, 2012). To address these issues and establish a practical, scalable, and economical protein interaction mapping method, we developed the Barcode Fusion Genetics (BFG) technology.

## Results

### Principle of Barcode Fusion Genetics

Modern sequencing-based phenotypic screening commonly uses DNA barcodes to identify single specific engineered changes—for example, gene deletions (Smith *et al*, 2009), or gene-targeting RNAi (Berns *et al*, 2004) or CRISPR reagents (Hart *et al*, 2015). However, it is frequently interesting, for example, for genetic interactions, to observe phenotypes resulting from two perturbations. The BFG technology enables phenotypic analysis of a heterogeneous pool of millions of yeast strains, each having two engineered loci or genes of interest. In BFG, a doubly engineered cell pool is prepared so that each of two engineered loci in each cell is associated with a distinct DNA barcode flanked by site-specific recombination sequences (Fig 1A). Once a given selection has been applied to the strain pool, barcodes representing different engineered changes are fused by site-specific recombination (Fig 1B). Strain abundances can then be quantified by amplification and deep sequencing of fused barcodes

(Fig 1B). Here, we apply the BFG concept for increased efficiency of yeast two-hybrid protein interaction screening.

### Barcode Fusion Genetics-Yeast Two-Hybrid (BFG-Y2H) technology

In preparation for BFG-Y2H, libraries of haploid strains are created, each carrying a plasmid that (i) expresses either a bait or prey protein and (ii) encodes a barcode locus carrying two strain-specific barcodes ("BC1" and "BC2", Fig EV1A). The directional site-specific *loxP* and *lox2272* recombination sites (Livet *et al*, 2007) flank the bait-BC1 and prey-BC2 barcodes (Fig EV1A). To screen for protein interactions, the bait and prey libraries are pooled and mated *en masse* to create a diploid cell population representing all protein combinations (Fig 2). Diploid cells corresponding to protein interactions are then enriched via selection for the expression of the Y2H reporter gene *HIS3*. The selected cells are then pooled and treated with doxycycline to induce Cre recombinase expression. Because Cre selectively recombines *loxP* with *loxP* and *lox2272* with *lox2272*, Cre induces a double-crossover event that physically swaps the bait-BC1 and prey-BC2 barcodes between the two plasmids. This leaves a chimeric bait–prey "BC1-BC1" barcode fusion on the prey plasmid and another chimeric bait–prey "BC2-BC2" barcode fusion on the bait plasmid (Appendix Note S1 and Fig EV2). After cell lysis and plasmid DNA extraction, the fused-barcode regions are amplified by PCR, and the interacting protein pairs are identified by massively parallel sequencing of the fused barcodes.

We derived two BFG-Y2H strains—"toolkit-**a**" and "toolkit-**α**"—from strains used successfully in recent large-scale Y2H studies (Yu *et al*, 2008, 2011; Simonis *et al*, 2009; Rolland *et al*, 2014; see Materials and Methods) to permit expression of Cre recombinase via the Tet-On system (Belli *et al*, 1998; Fig EV1). The toolkit-**a** strain expresses *rtTA* (needed for the Tet-On system) and serves to host barcoded prey plasmids. The toolkit-**α** strain encodes *Cre* under control of the *tetO₂* promoter and hosts barcoded bait plasmids. We evaluated mixed cell populations and found that barcode swapping between plasmids occurred specifically within but not between cells, demonstrating *in vivo* barcode fusion (Appendix Note S1 and Fig EV2).

### A BFG-Y2H screen targeting human centrosomal proteins

As a proof-of-principle study, we screened a "CENT" matrix, including human centrosomal and centrosome-associated proteins, curated from a combination of sources (Andersen *et al*, 2003; Lamesch *et al*, 2007; Temple *et al*, 2009), encompassing 143 bait and 162 prey ORFs (Table EV1). Barcoded haploid strains were generated by in-yeast assembly (Ma *et al*, 1987; Gibson, 2009) to each carry a uniquely barcoded plasmid expressing a specific bait or prey (see Materials and Methods). Four DNA fragments were pooled and co-transformed into yeast with overlapping sequences to be assembled by homologous recombination (Fig 3A). These four fragments correspond to the following: (i) a bait or prey-encoding ORF; (ii) either the DNA-binding or activation domain of Gal4; (iii) a barcode locus; and (iv) the plasmid "backbone", which encodes a marker enabling selection for cells with correctly assembled BFG-Y2H plasmids (Fig 3B). Among the 161 centrosome-related ORFs attempted (Table EV1), 112 bait and 131 prey ORF fragments passed stringent quality controls both by Sanger sequencing and by PCR

**A**

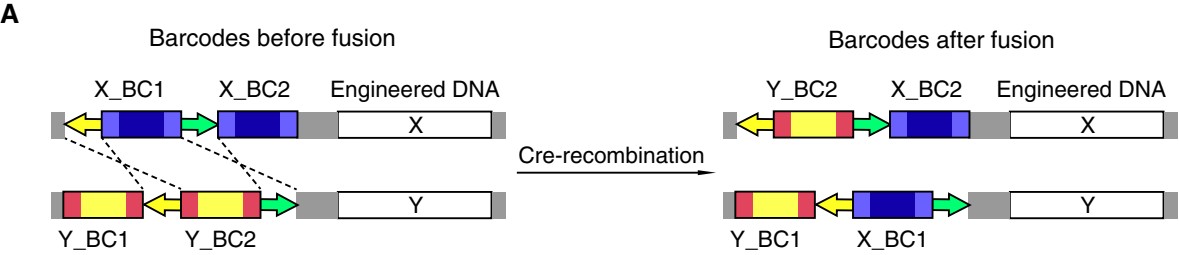

**B**

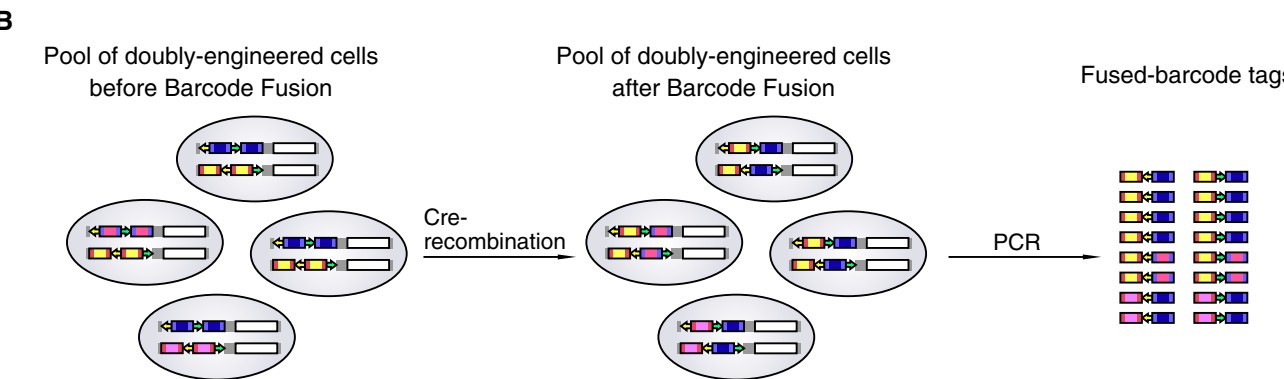

**Figure 1. Principle of Barcode Fusion Genetics, a technology to generate fused barcodes that uniquely identify the presence of a specific combination of engineered loci.**

A Each cell carries two engineered loci, such that each locus is identified by the presence of a barcode flanked by site-specific recombination sites. In the presence of Cre recombinase, a double-crossover DNA recombination is induced to form chimeric "fused" barcodes that represent the combination of loci.

B Multiple pairwise combinations of reagents can be tested in a pool. Fused barcodes can be amplified and analyzed by deep sequencing to analyze the abundance of cells corresponding to each X-Y combination.

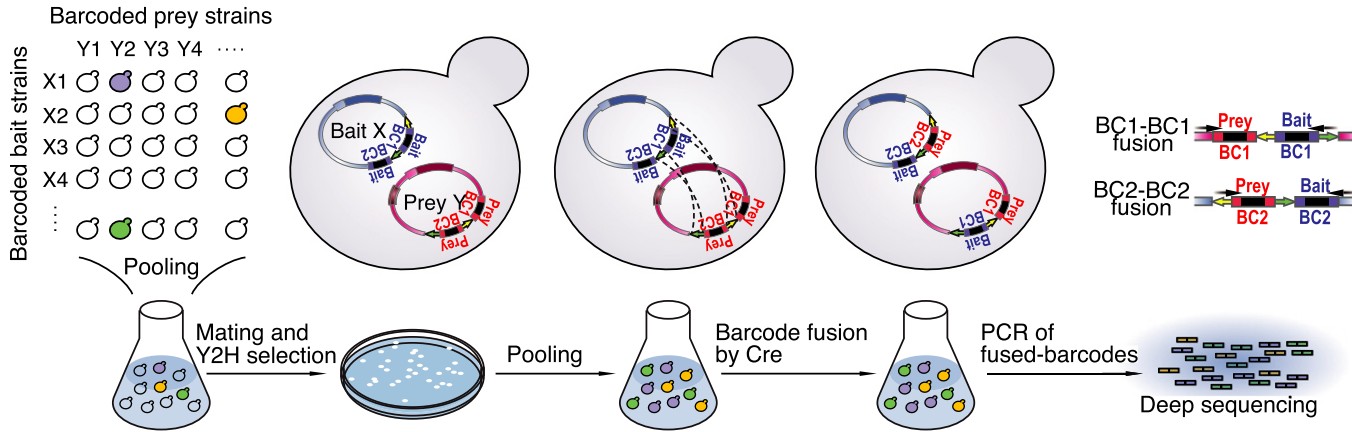

**Figure 2. Design of the BFG-Y2H technology.**

A pool of diploid cells, potentially expressing all possible pairwise combinations of bait and prey fusion proteins, is generated via *en masse* yeast mating, in which a haploid pool of bait strains (MATα) is mated with a pool of prey strains (MATa). Diploid cells surviving the Y2H selection are pooled, and Cre recombinase is induced to swap the positions of the bait-BC1 and prey-BC2 and to generate chimeric BC1-BC1 and BC2-BC2 barcode fusions that each uniquely identifies a candidate X-Y interaction. Cells are then lysed, plasmids are extracted, and a DNA sequencing library is prepared by PCR for both BC1-BC1 and BC2-BC2 fused barcodes. Finally, protein interactions are identified according to the enrichment of sequencing read counts for fused barcodes corresponding to particular protein pairs.

amplicon length. We also included 31 calibration set protein pairs (see Materials and Methods). For each bait or prey ORF, we prepared two uniquely barcoded strains (286 bait strains and 324

prey strains in total). To assess the quality of this assembly protocol, we isolated 23 single colonies from each of six assembly reactions containing different ORFs (*LCP2*, *FKBP3*, and *BDNF* for bait strain

 

generation and *NCK1*, *NQO2*, and *NTF5* for barcoded prey) and analyzed these by PCR (Fig 3C). Of the 138 colonies evaluated, only two (1.4%) showed misassembled plasmids (both from *LCP2*, which showed success in the majority of its colonies), demonstrating high fidelity of the in-yeast assembly process.

To generate the barcode locus fragments used in the above procedure, we used Gibson *in vitro* DNA assembly (Gibson *et al*, 2009) of synthetic dsDNA-containing random 25-bp degenerate regions (Fig EV3). Each successful assembly joined BC1, BC2, *loxP*, and *lox2272* sites in the appropriate order, together with a backbone DNA fragment of a Gateway LR cloning-compatible destination plasmid (Walhout *et al*, 2000). Thus, each resulting "barcode carrier" plasmid contained a randomized sequence at each of two barcode positions. The assembled DNA sample was transformed into competent *E. coli* cells, such that each transformant carried a single randomly barcoded destination plasmid. Colonies were picked and arrayed into 384-well plates. To identify the barcode in each well position within the stack of plates, a row-column-plate-indexed PCR (RCP-PCR) generated amplicons that each contained a barcode locus flanked by sequence indices identifying plate and well position (Appendix Note S2 and Fig EV3). RCP-PCR amplicons were pooled and subjected to next-generation sequencing to identify barcode and index sequences. Barcode fragments were later amplified directly from re-arrayed *E. coli* strains. To date, we have created a reusable Gateway destination collection of more than 2,500 bait and 2,500 prey barcode carrier plasmids.

To initiate the BFG-Y2H screen, bait and prey strains were individually grown to saturation in 96-deep-well plates to minimize variation of strain abundance within pools. Strains were pooled, pools were mated, and diploids were selected. Mated diploid cells were spread on agar plates with: (i) non-selective control media containing excess histidine ("+His"), (ii) Y2H-selective media ("–His"), and (iii) stringent Y2H-selective media lacking histidine and supplemented by amino-1,2,4-triazole ("3-AT"), a competitive inhibitor of the *HIS3* gene product (see BFG-Y2H procedure in Materials and Methods for detailed media descriptions). The experimental scale was designed to achieve an average of ~100 DNA molecules representing each distinct protein pair at the most restrictive population bottleneck (yeast plasmid extraction for non-selective +His condition), as supported by a computational Monte Carlo simulation of the entire screening process (Appendix Note S3 and Appendix Fig S1).

As expected given that the vast majority of protein pairs do not interact, count distributions from selective media were sparser than those of non-selective condition (Figs 4A and B, and EV4) and therefore exhibited a higher dynamic range because high counts were observed for a relatively small number of protein pairs surviving the Y2H selection. High-background levels were observed for some bait proteins under Y2H-selective conditions (Fig 4C), suggesting that these proteins have some ability to "auto-activate" the reporter gene by recruiting RNA polymerase II directly in the absence of an interacting prey protein. Fused barcodes corresponding to seven of such "auto-activators", *GMNN*, *HAUS6*, *HAUS8*, *NIN*, *PPP2R3C*, *YWHAE*, and *YWHAG*, were highly abundant in the selective conditions of the initial screen. To provide biological replication and to assess the effect of high-background baits on the screen, we

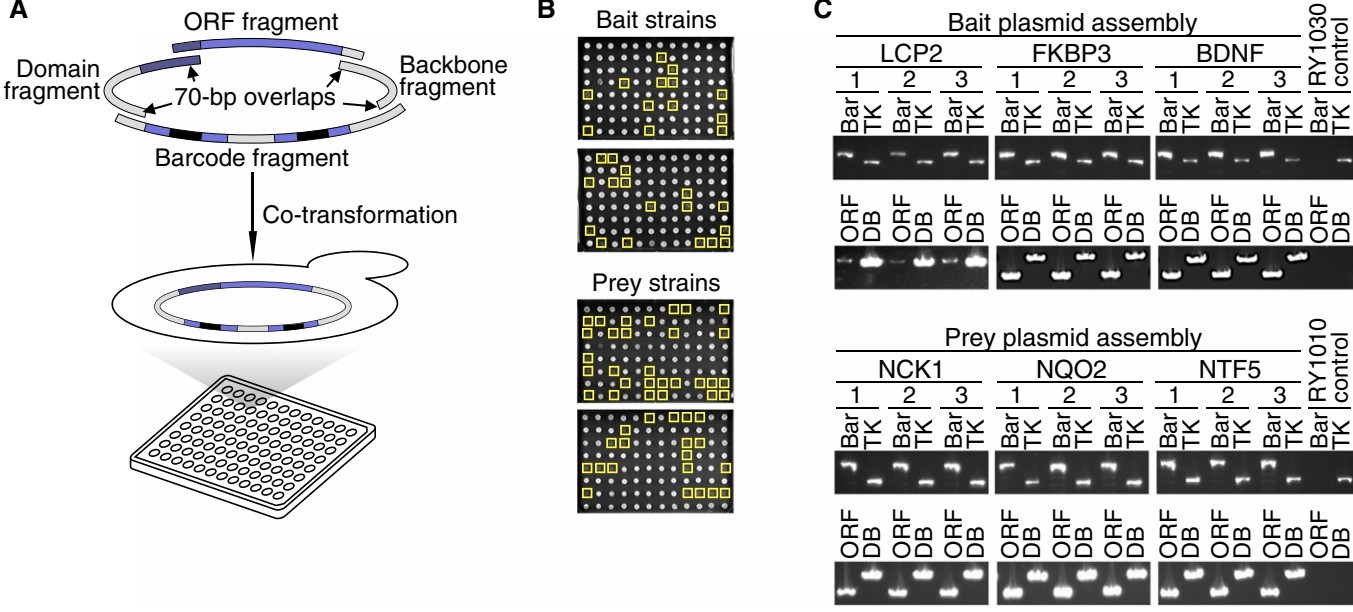

**Figure 3.  Massively parallel generation of barcoded Y2H strains.**

A   Library-scale in-yeast assembly to generate Y2H strains carrying barcoded ORF-expressing plasmids. In each reaction, the Gal4 DNA binding or activation domain, and ORF, barcode and backbone DNA fragments were directly assembled *in vivo* in either the toolkit-**a** or toolkit-**α** strain background.

B   Barcoded Y2H strains derived by in-yeast assembly. Colony growth indicates yeast cells harboring the correctly assembled plasmids. The yellow boxes denote "no ORF fragment" negative controls.

C   Quality confirmation of in-yeast assembly-based barcoded strain generation. After in-yeast assembly, single colonies were isolated and the DNA fragments were recovered by yeast colony PCR. "TK" denotes genotyping PCR to confirm the presence of the chromosomal locus that defines the toolkit strains.

removed the auto-activator strains and repeated the screen (Fig EV4).

There was a high correlation between the average BC1-BC1 and BC2-BC2 fused-barcode counts corresponding to a given protein pair under selective (−His) conditions ($R = 0.90$; $P < 10^{-15}$; Fig 4D). This indicates that read counts of fused barcodes primarily reflect the makeup of the cell populations without major barcode-specific effects arising from differences in efficiency of PCR or sequencing. There was also a high correlation of average fused-barcode counts between strain replicates that express the same bait and prey proteins, but which did not share same barcodes ($R = 0.70$; $P < 10^{-15}$; Fig 4D), suggesting not only that the screening method is robust, but also that the screen was close to saturation and the sampling sensitivity was high. The seven auto-activator bait strains did not dominate the selective

condition screens and the correlation between two screening variants (with and without auto-activators) was also high ($R = 0.85$; $P < 10^{-15}$; Fig 4D).

To assess the efficiency of Cre-mediated "swapping" of barcodes between plasmids, we examined the pool of plasmids isolated after Cre induction and cell lysis of the CENT screen, but prior to PCR amplification of the barcode loci. This DNA was sequenced *en masse* using the Illumina Nextera method. By assessing the relative abundance of original and chimeric barcode loci, we estimated 16–27% of cells from each Y2H-positive colony to yield fused barcodes after overnight Cre induction (Figs 4E and EV5).

Accurately estimating the effects of Y2H selection on each protein pair must account for several factors: (i) uneven strain abundance in the initial bait and prey; (ii) potential competitive growth effects of bait and prey expression; and (iii) slight barcode-dependent

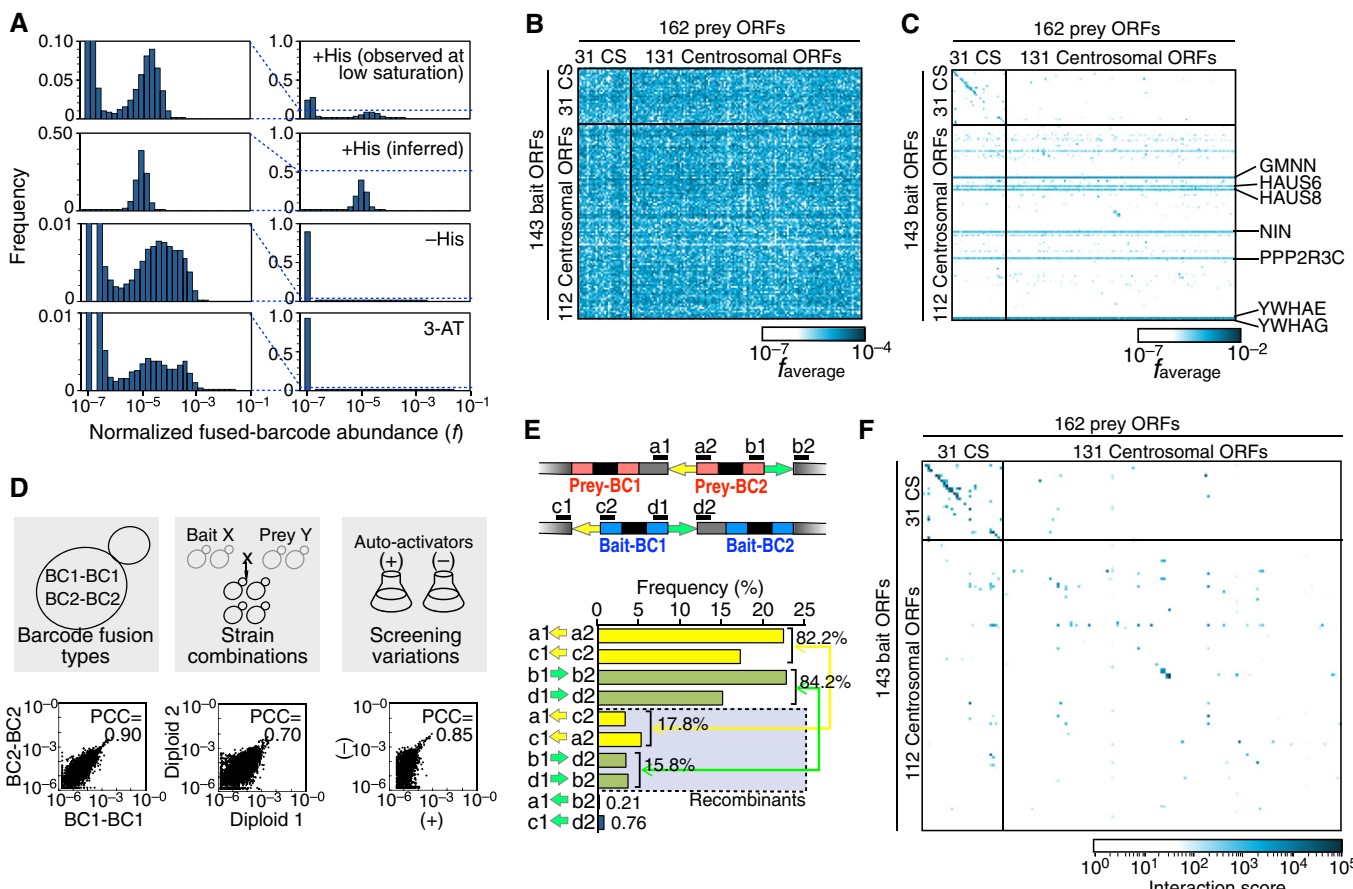

**Figure 4.  Screening coverage, reproducibility, and other features of BFG-Y2H CENT screen.**

A     Normalized fused-barcode abundance is shown for 1) non-selective conditions, based on observed fused-barcode abundance at a sequencing depth that is only sufficient for accurately determining barcode marginal abundance ("+His observed at low saturation"); 2) non-selective conditions, as inferred from marginal abundance of single-barcode frequencies ("+His inferred"), and 3) selective conditions based on observed fused-barcode abundance ("−His" and "3-AT").

B, C  Average of normalized fused-barcode count for each ORF pair ($f_{average}$) in (B) the non-selective (+His) condition and (C) in the selective (−His) condition. CS: calibration set space spiked in the screen.

D     Correlation of $f_{average}$ between different pairs of replicate types in the selective conditions (scatter plots are log-scale).

E     Analysis of barcode fusion efficiency. Frequencies of 7-bp flanking motif combinations located upstream and downstream of *loxP* (yellow arrow) or *lox2272* (green arrow) sites were analyzed by Illumina Nextera sequencing for the −His condition.

F     Interaction score profile for the CENT screen with parameters optimized according to the Matthews correlation coefficient (MCC) to recapture previously reported Y2H interactions.

differences in amplification and sequencing. We therefore rescaled read counts in the –His and 3-AT conditions using the total row or column counts of each bait and prey strain in the +His condition matrix. The normalized barcode count $s$ was further rescaled to account for differences in background auto-activation activity to yield an interaction signal $s'$ for each barcode pair (Appendix Note S4). For each protein pair, the two BFG-Y2H screens produced a total of 32 $s'$ signal measurements: two fused-barcode variants (BC1-BC1 and BC2-BC2) × four diploid replicates × two selective conditions × two library variants (with and without the seven auto-activators). The best method for deriving a single interaction score from replicates was optimized according to the ability to recapture previously reported Y2H-positive protein pairs with the best balance of precision and recall according to the Matthews correlation coefficient (MCC; Guda *et al*, 2004). Although 3-AT is commonly used in Y2H screening to increase the dependence of growth on *HIS3* expression and thus to screen for interactions with higher stringency (Dreze *et al*, 2010), using only the "–His" screen here yielded increased sensitivity and the best overall performance (MCC = 0.52 as opposed to MCC = 0.39 for the 3-AT screen; Appendix Fig S2). Protein pairs achieving a high interaction score showed a clear enrichment for known interactions (Fig 5A and B).

### A BFG-Y2H screen targeting cancer and cell cycle-related proteins

To demonstrate BFG-Y2H at larger scale and broaden discovery of new interactions, we next carried out BFG-Y2H screening on a "CCC" matrix, which includes proteins implicated in cancer according to the COSMIC database (Forbes *et al*, 2015), as well as cyclins, cell cycle kinases, and cell cycle kinase substrates (Hornbeck *et al*, 2012), encompassing 392 bait and 377 prey doubly barcoded ORFs (Table EV1). CCC contained sixfold more protein pairs than the CENT matrix. Screening and scoring of CCC was carried out in essentially the same way as for CENT, except for small differences in the in-yeast assembly procedure (Materials and Methods and Appendix Note S4).

### Using the CENT and CCC screens to evaluate BFG-Y2H performance

Like CENT (Fig 5A), protein pairs in the CCC screen with high interaction scores were enriched for previously known interactions (Appendix Fig S3). The best performance of recapturing known interactions (in terms of the MCC performance measure) was achieved by taking the top 55 and 54 protein pairs for CENT and CCC screens, respectively. Pairwise retests of the top 100 protein pairs for the CENT and CCC screens and arbitrarily chosen low-scoring pairs were performed using an independent set of non-barcoded Y2H strains. For each protein pair, ORFs were transferred from Gateway entry plasmids to the non-barcoded bait and prey destination plasmids by Gateway LR cloning. Retests were performed both in the BFG-Y2H toolkit strains and in the parental "Y-strain" backgrounds described previously (James *et al*, 1996). Both retest-positive pairs and previously known interactions were enriched at higher interaction scores. To assess the precision of interactions emerging from the primary screen, we first examined interactions above the MCC-optimal rank (55 and 54 pairs for CENT and CCC,

respectively). After excluding auto-activators and untested pairs, pairwise retests verified 23/47 (49%) of primary "hits" from CENT and 25/46 (54%) from the CCC screen including 4/11 (36%) and 5/14 (36%) novel pairs, respectively. Within the top 100 protein pairs, 32/82 (39%) and 36/66 (55%) pairs were verified, with verification for 10/42 (24%) and 13/30 (43%) novel pairs (Fig 5A, Table EV2 and Appendix Fig S3). By contrast, only one pair (1.2%) of the 84 tested BFG-Y2H negatives was verified by pairwise Y2H retesting (Table EV2). A retest-positive rate of ~50% from the primary BFG-Y2H screen is within the normal range for the current Y2H pipeline at CCSB (Rual *et al*, 2005; Yu *et al*, 2008; Simonis *et al*, 2009).

Despite the fact that many of the pairs we tested (87% and 78% in CENT and CCC, respectively) had been previously screened (Rolland *et al*, 2014), BFG-Y2H uncovered several interesting new interactions. For example, the CCNDBP1 protein, which is known to negatively regulate cell cycle progression and to have tumor suppressor functions (Ma *et al*, 2007), was found to physically interact with TFPT, which is involved in DNA repair and promotes apoptosis in a p53-independent manner (Franchini *et al*, 2006). This interaction might play a role in the regulation of cell death during cell cycle progression. BFG-Y2H also identified an interaction between RBPMS, a transcriptional co-activator with a role in TGF beta signaling (Sun *et al*, 2006) and SMAD3, a transcription factor of the TGF beta pathway. This interaction was only recently reported to be responsible for cell growth and migration inhibition in breast cancer cell lines (Fu *et al*, 2015). Other RBPMS partners we identified included PICALM, a clathrin adaptor with a role in Alzheimer's disease (Moreau *et al*, 2014), TCF7L2, a Wnt signaling transcription factor (Korinek *et al*, 1997) and PATZ1, a transcriptional regulator with a role in cell death and proliferation (Valentino *et al*, 2013) and differentiation (Ow *et al*, 2014). There is a very well established role of the TGF beta pathway in cell proliferation, cancer, and Alzheimer's disease (Kajdaniuk *et al*, 2013) and these interactions can contribute to improved understanding of the TGF beta pathway and its roles.

### Validation of BFG-Y2H interactions by calibrated orthogonal assays

The CENT and CCC screen results were validated using an orthogonal *Gaussia princeps* luciferase protein complementation assay (GPCA) in human cells (Remy & Michnick, 2006; Cassonnet *et al*, 2011). Among the top 100 hits of each of the CENT and CCC screens, GPCA vectors were successfully generated for 83 and 64 protein pairs, respectively (Fig 5A and Appendix Fig S3).

Among the top 55 BFG-Y2H hits in the CENT screen, 74% of the 46 tested were validated by GPCA. In the top 100 of CENT, 48% of 83 tested were GPCA positive. By contrast, of 54 BFG-Y2H-negative pairs from CENT that were tested by GPCA, only 2 (4%) were GPCA positive. From the CCC screen, 46% of 41 GPCA-tested pairs in the top 54 pairs and 39% of 72 GPCA-tested pairs among the top 100 pairs were GPCA positive. By contrast, only 1 (3%) of 36 BFG-Y2H-negative pairs were GPCA positive. The validation rate of BFG-Y2H hits by GPCA compared well with that of a state-of-the-art Y2H study (Sahni *et al*, 2015), in which GPCA validated 59% of 165 Y2H-positive pairs and 41% of the 17 Y2H negatives. In another state-of-the-art Y2H study (Hill *et al*, 2014), GPCA validated ~35% of Y2H hits and 0% of Y2H-negative pairs that were examined.

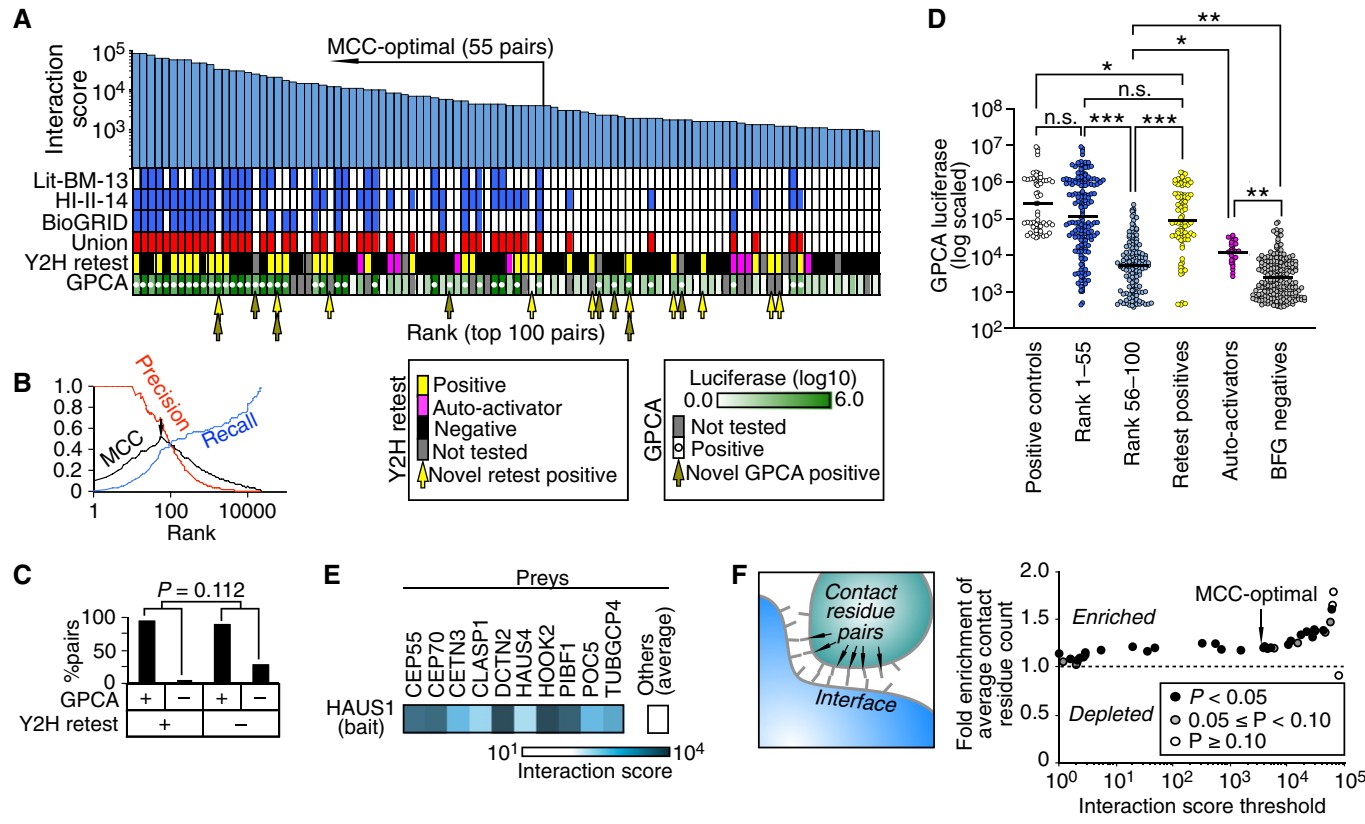

**Figure 5. BFG-Y2H efficiently captures protein interactions.**

A   Top 100 protein pairs scored by BFG-Y2H, and their presence in a high-quality literature-curated protein interaction set (Lit-BM-13), a recent systematic high-quality human interactome dataset (HI-II-14), or the curated BioGRID protein interaction dataset (see Materials and Methods). "Union" represents the union of interacting protein pairs in Lit-BM-13, HI-II-14, and BioGRID.

B   Performance in recovering previously reported interactions ("Union").

C   Recovery rate by GPCA for BFG-Y2H-positive (+) versus BFG-Y2H-negative (−) hits and pairwise retest-positive (+) versus retest-negative (−) hits.

D   Distribution of GPCA luciferase intensities (quadruplicates) for protein pairs in the positive control (defined as the overlap of the GPCA-tested space with the union of the HI-II-14 and Lit-BM-13 datasets, Rolland et al, 2014; Table EV2), rank 1–55, 56–100, pairwise Y2H retested positives, auto-activators in the pairwise Y2H pipeline and BFG-Y2H negative pairs. *$P < 0.05$, **$P < 10^{-5}$, and ***$P < 10^{-15}$ (Mann–Whitney U-test).

E   HAUS1 hits captured by BFG-Y2H.

F   Fold enrichment of residue contacts at protein interfaces for different interaction score thresholds. Fold-change is calculated as the ratio of the average number of residue contacts for the two groups of protein pairs separated by each interaction score threshold. P-value was calculated using the Mann–Whitney U-test.

Interestingly, the GPCA validation rate was not significantly different for the subset of BFG-Y2H-positive hits that showed positive Y2H retests (Fig 5C). This suggests that filtering by pairwise Y2H diminished the output of our BFG-Y2H screens without substantially improving the quality of the results. We also found that our interaction score correlated with GPCA signal intensity (Figs 5D and EV4), highlighting the reproducibility of our interaction pairs in an orthogonal assay. The GPCA luciferase signal has been reported to correlate with protein interaction affinity (Szklarczyk et al, 2015), so that a quantitative correlation with GPCA suggests BFG-Y2H's potential for measuring interaction strength.

## BFG-Y2H can detect interactions for high-background "auto-activator" baits

Some bait proteins are "auto-activators" that can, to varying degrees, independently recruit RNA polymerase II and thus activate the reporter gene. Current Y2H pipelines typically remove auto-activators prior to large-scale screening. By contrast, BFG-Y2H can identify interactions between protein pairs involving auto-activators. Specifically, protein pairs that were within the top 100 and scored as auto-activators by Y2H pairwise retest yielded significantly higher GPCA signal intensities than protein pairs scored negatively by BFG-Y2H ($P < 10^{-9}$; Fig 5D). For example, the TACC3 homodimer interaction, which ranked 63rd in the CCC screen, was successfully validated by GPCA (Table EV2) despite behaving as a Y2H auto-activator in pairwise retesting. Other examples included human Augmin complex (HAUS) subunit proteins of the CENT screen. HAUS6 and 8 showed strong auto-activation (Fig 4C). HAUS1, 3, 4, and 7 showed weak auto-activation with HAUS1 and 7 also being detected as auto-activators in pairwise retesting (Table EV2). Nevertheless, several HAUS1 connections to centrosome-related proteins emerged with BFG-Y2H scores above background (Fig 5E). The Augmin complex regulates

centrosome and spindle assembly, co-localizing with the centrosome during interphase but with microtubules during mitosis (Lawo *et al*, 2009). In the CENT screen, HAUS1 had a markedly higher interaction score with proteins localizing with centrioles or pericentriolar material (CEP55, CEP70, and POC5) and proteins promoting microtubule dynamics (CEP70, CETN3, and CLASP1). The mitotic co-localization of HAUS1 with microtubules might be explained by its interactions with dynactin (DCTN2), which anchors microtubules to the centrosome (Echeverri *et al*, 1996) and interacts with microtubule regulator MAPRE3 (Berrueta *et al*, 1999). The BFG-Y2H screens also captured the interaction between DCTN2 and MAPRE3. HAUS1 had a markedly higher interaction score with the gamma-tubulin ring complex (TuRC) protein TUBGCP4 than its other partners, which is consistent with previous observations that the Augmin complex is required for localization of gamma-TuRC (and chromosome segregation and cytokinesis) in human cells (Uehara *et al*, 2009). We also found that HAUS1 interacted with HOOK2, which is known to interact with microtubules (Szebenyi *et al*, 2007; Fig 5E).

### High-scoring BFG-Y2H interactions show increased interfacial contacts in co-crystal structures

To investigate whether the quantitative interaction score resulting from BFG-Y2H correlates with interaction strength, we calculated the number of residue–residue contacts in the three-dimensional protein interface where co-crystal structures or high-quality homology models were available. We found that protein pairs above the MCC-optimal interaction score threshold have significantly more interfacial residue–residue contacts (Fig 5F and Table EV3). These results support the idea that quantitative scores from BFG-Y2H have potential value in estimating protein interaction strength, although this remains to be confirmed.

### A more efficient *en masse* strategy for producing barcoded plasmids

To facilitate larger-scale BFG-Y2H screens, we extended a previously described "pooled ORF expression technology" (Waybright *et al*, 2008) to more efficiently produce pools of barcoded bait and prey plasmids. In this strategy, here termed *en masse* recombinational cloning reaction, two pools of vectors are combined: (i) A pool of entry clones corresponding to a collection of ORFs; (ii) a highly complex pool of randomly barcoded bait or prey destination plasmids. The Gateway LR Clonase recombinational cloning reaction is then carried out *en masse* to produce a pool of barcoded bait and prey expression plasmids (Fig 6A). For each reaction pool, bacterial cells are then transformed to obtain clonal colonies of barcoded bait or prey expression plasmids. Bacterial colonies are robotically picked, arrayed in 384-well plates, and sequenced to identify the barcode sequence and ORF of each clone. A subset of sequence-identified bacterial strains is chosen to obtain a similar number of barcoded plasmid versions for each ORF. The chosen strains are robotically re-arrayed, grown overnight, then pooled for a single plasmid DNA purification (Fig 6B). Barcoded bait or prey pools are used to transform toolkit-**α** or toolkit-**a** strains, respectively, to generate BFG-Y2H-ready strain pools.

### Application of the *en masse* recombinational cloning strategy to efficiently generate barcoded plasmid sets

We applied the *en masse* recombinational cloning approach to two target ORF spaces—"CV" and "A" (Materials and Methods and Table EV1). The first space CV corresponded to the union of 218 COSMIC ORFs, 272 human "virhostome" proteins targeted by DNA tumor viral proteins (Rozenblatt-Rosen *et al*, 2012), and a set of 381 arbitrary picked ORFs including calibration ORF pairs, for a total of 767 unique ORFs. Two A pools, each of which was composed of 1,896 arbitrarily picked ORFs, were, respectively, used to generate barcoded bait and prey libraries. The successfully barcoded bait and prey libraries were used for two interactome screens: CV described above, and "CVA", a larger screening space combining both CV and A libraries, to demonstrate scalability of BFG-Y2H.

From each of the two *en masse* Gateway reactions, bacterial transformant colonies were picked and arrayed to 384-well LB+ampicillin plates. Among high-quality clones identified by sequencing, the number of different barcodes assigned to each ORF varied (Fig 6C). From each set, we selected and re-arrayed high-quality bacterial clones, so that each ORF was assigned a maximum of four different barcodes. Out of the 767 CV ORFs subjected to the *en masse* Gateway reaction, 623 (81%) and 619 (81%) were, respectively, recovered as barcoded bait clones and prey clones. Out of the 1,896 A ORFs, 1,169 (62%) and 1,208 (64%) were obtained as barcoded bait clones and prey clones, respectively (Fig 6D). Note that the probability of obtaining a barcoded clone for any given ORF depends on the number of bacterial colonies obtained after the *en masse* Gateway reaction relative to the number of input ORFs. After generating barcoded haploid yeast strain pools for each target ORF set, we created bait and prey strain pools corresponding to both CV and CVA screens.

### *En masse* Gateway-based BFG-Y2H screening

The BFG-Y2H process for CV and CVA was similar to that of the CENT and CCC screens. Because heterogeneity was higher in the haploid pool of the *en masse* Gateway reaction-based screen than in the in-yeast assembly-based screen, we considered only strains exhibiting sufficient barcode counts in the unselected (+His) pool (Appendix Note S4). Two replicate screens were performed for CVA in parallel and *s'* signals were combined to calculate interaction scores. Of the pool of barcoded expression plasmids transformed into BFG-Y2H toolkit strains for the CV screen, 578 (93%) of bait ORFs and 579 (94%) of prey ORFs (334,662 pairs) were scored. For the CVA screen, 1,571 (88%) of bait ORFs and 1,639 (90%) of prey ORFs (totaling 2.6 million pairs) were scored (Figs 6D and 7A and Table EV1). ORF attrition occurred primarily at the *en masse* Gateway reaction stage for overall slightly longer ORFs (Fig 6D; no marked bias throughout the pooled yeast transformation), suggesting an iterative strategy to map a given space: (i) carry out a first round *en masse* Gateway reaction as described above; (ii) carry out a second iteration for bait and prey ORFs lost at any prior stage; (iii) pool barcoded bait and prey clones and carry out BFG-Y2H. With single rounds of *en masse* Gateway reaction and BFG-Y2H, we obtained 389 and 591 interacting protein pairs in the CV and CVA screens, respectively, at the MCC-optimal interaction score thresholds recapturing previously known Y2H hits (Fig 7B and Table EV2).

                                                                    

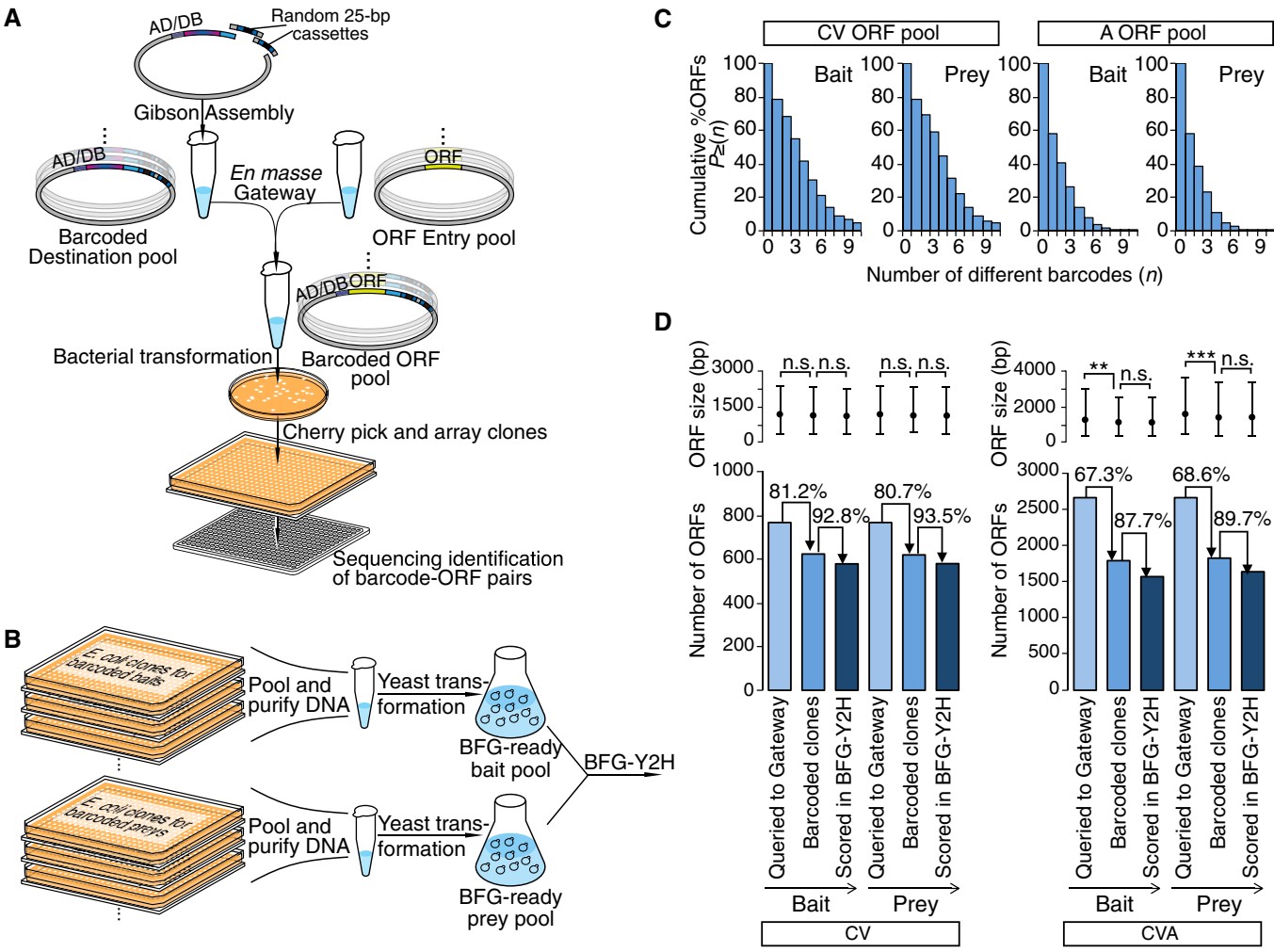

**Figure 6. Scalable generation of barcoded bait and prey strains based on a pooled recombinational cloning reaction.**

A  Schematic representation of the *en masse* recombinational cloning process. Randomly barcoded bait or prey destination plasmid pool was combined with a pool of entry ORF plasmids and subjected to a Gateway LR reaction. Randomly barcoded ORF expression clones were isolated by bacterial transformation and colony picking and identified by sequencing.

B  Generation of BFG-Y2H-ready bait and prey haploid pools by *en masse* transformation of purified barcoded bait and prey expression plasmid pools to the appropriate mating type yeast cells.

C  Fraction of ORFs assigned to at least *n* barcodes indicated on the horizontal axis.

D  Attrition of ORFs and their lengths at steps of the *en masse* recombinational cloning-based BFG-Y2H procedure. **$P < 10^{-4}$ and ***$P < 10^{-7}$.

To enable comparison across screens, we included ORFs corresponding to the common set of 31 protein pairs described above, of which 18 yielded a score in all of the four BFG-Y2H screens. Of the 18 pairs in this "calibration set", we found 13, 11, 13, and 11 interactions in CENT, CCC, CV, and CVA screens, respectively (Fig 7C), indicating consistent sensitivity at increasing scales. Furthermore, within the sub-matrix of protein pairs examined in both the CV and CVA screens (312 thousand pairs), the resulting interactions overlapped significantly ($P < 10^{-15}$; Fig 7D). Thus, BFG-Y2H is scalable to matrices of at least ~2.5 M protein pairs with no increase in hands-on time during the screen. The costs of sequencing scale linearly with matrix size, while costs relating to barcoded strain construction scale as the square root of matrix size.

**Performance comparison between BFG-Y2H and state-of-the-art Y2H**

To compare the practical performance of BFG-Y2H with state-of-the-art Y2H methods, the four BFG-Y2H screen results and a recent high-quality Y2H-based human interactome dataset HI-II-14 (Rolland *et al*, 2014) were each evaluated using Lit-BM-13, a high-quality set of literature-curated physical interactions (Rolland *et al*, 2014), as a benchmark standard. Each screen was assessed according to its ability to recover the subset of Lit-BM-13 falling within its tested space (Fig 7E). Generally, BFG-Y2H screening outperformed state-of-the-art Y2H in terms of precision albeit with reduced recall (Fig 7F). According to the MCC measure, which balances precision and recall, BFG-Y2H performed on par with

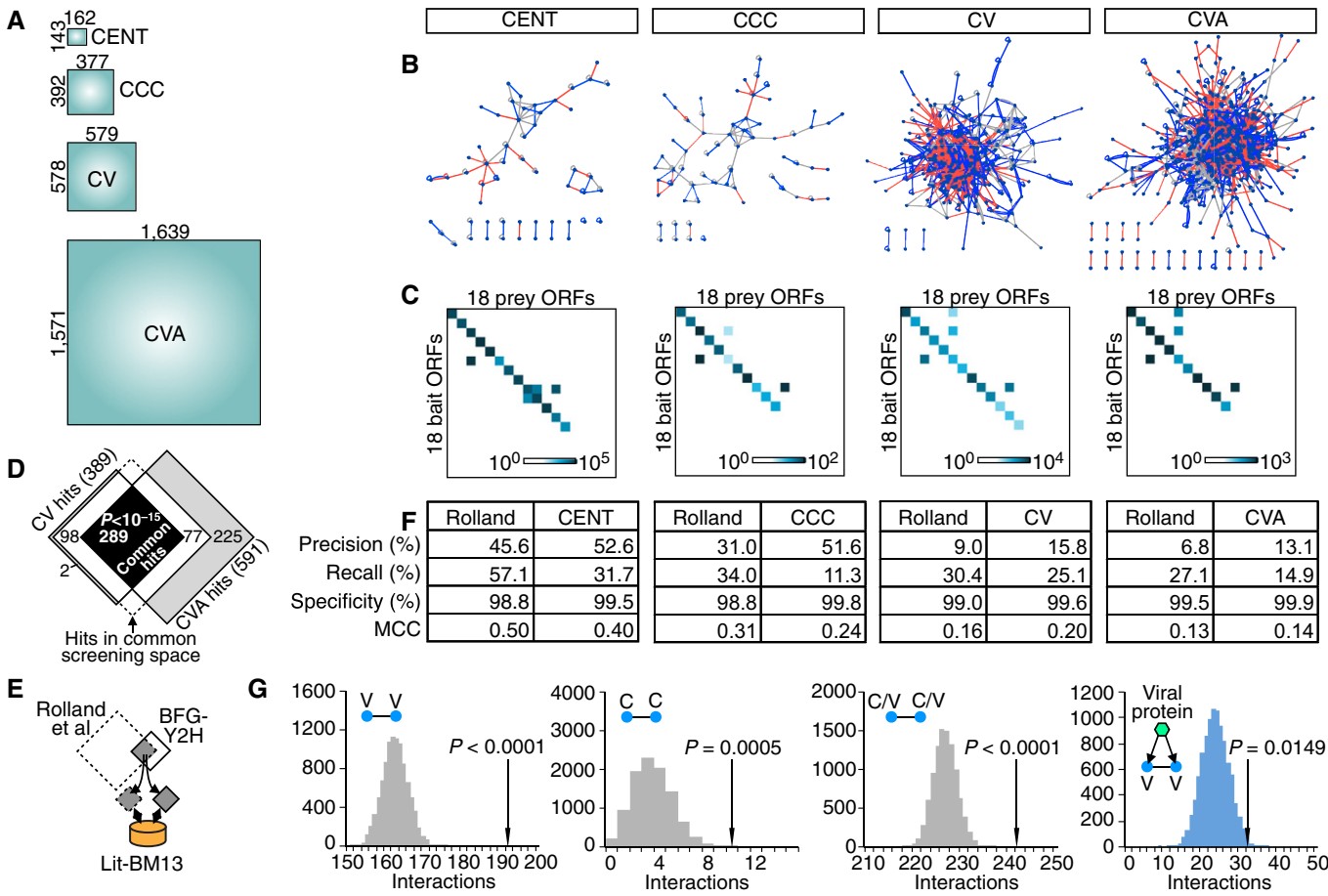

**Figure 7. Scalability and performance of BFG-Y2H.**

A Schematic representation of the increasing size of the four protein pair spaces tested (CENT, CCC, CV, and CVA).

B Protein interaction networks identified by each BFG-Y2H screen. Red lines indicate novel interactions, blue lines indicate previously known interactions (those in the "Union" set) captured by BFG-Y2H, and gray lines denote known interactions among proteins in the hit list that were not captured by BFG-Y2H.

C Sub-matrices for the 18 calibration pairs that were commonly tested in all of the four screens. The X and Y ORFs were ordered to present calibration pairs on the diagonal.

D Overlap between CV and CVA interactions.

E, F The performance of each BFG-Y2H screen was measured using Lit-BM-13 and compared with that of HI-II-14 after restricting both screening spaces to their common ORFs.

G Numbers of protein interactions among virhostome proteins (V-V) and among COSMIC cancer proteins (C-C) and number of virhostome interactions targeted by same viral proteins. Gray bars demonstrate expectations from the randomly generated networks by a random edge rewiring.

current Y2H. Specifically, the larger *en masse* recombinational-cloning-based BFG-Y2H screens showed MCC = 0.20 and 0.14 for CV and CVA, respectively, as compared with 0.16 and 0.13 for the HI-II-14 dataset within the corresponding BFG-Y2H search spaces. The smaller in-yeast assembly-based BFG-Y2H screens CENT and CCC yielded MCC = 0.40 and 0.24, respectively, as compared with 0.50 and 0.31 for the HI-II-14 dataset in the corresponding search spaces. It should be noted that HI-II-14 was based on the union of two primary screens and was filtered by pairwise retesting, while the performance of BFG-Y2H was based on single primary screens that were not filtered by secondary Y2H retesting.

## The CV protein network

The CV protein interaction network was composed of 389 interactions including 247 novel interactions (Fig 7B). After excluding

homodimeric interactions, there was significant enrichment for protein interactions between virhostome proteins, between COSMIC cancer proteins, and between virhostome and COSMIC cancer proteins, relative to networks generated by a random edge rewiring (Yachie et al, 2011). This supports the biological relevance of BFG-Y2H interactions.

We similarly found enrichment for interactions between pairs of human proteins where each is known to interact with a common viral protein (Fig 7G). As one example, Epstein–Barr virus (EBV)-targeting proteins DVL2 and KLHL12 were found to interact with each other. Wnt signaling activation and β-catenin accumulation are observed in EBV-infected B cells with type III latency expression and in the epithelial cells of the EBV-associated malignancy carcinoma (Hayward et al, 2006). DVL2 participates in Wnt signaling by binding to the cytoplasmic C-terminus of frizzled family membranes and transduces the Wnt signal downstream (Katoh, 2005). KLHL12

is a substrate-specific adapter of a BCR E3 ubiquitin ligase, which negatively regulates Wnt signaling by mediating ubiquitination and subsequent proteolysis of DVL3 (Angers *et al*, 2006). The newly discovered DVL2-KLHL12 interaction suggests the possibility that, like DVL3, DVL2 is regulated by KLHL12.

## Discussion

Although much has been learned from large-scale protein interaction mapping, it remains substantially incomplete in humans and all other organisms. Given limitations in assay sensitivity, completion will be asymptotic, requiring the use of multiple assay variants (Uetz *et al*, 2000). Screening matrices will grow as the square of the number of clones, with more clones needed to cover splice isoforms (Corominas *et al*, 2014) and other coding sequence variants (Sahni *et al*, 2015). Further adding to the dimension of the search space, it is now clear that screening with fragments can increase the assay sensitivity (Boxem *et al*, 2008), for example, by eliminating repressive domains. While significant improvements have been made to reduce the cost and effort required to screen large libraries of genes, state-of-the-art Y2H methods still require manual colony picking and multiple PCR steps for each Y2H-positive colony (Yu *et al*, 2011; Rolland *et al*, 2014). BFG-Y2H allows high-throughput Y2H screening at substantially lower cost and effort than the other Y2H procedures (Appendix Fig S4 and Table EV4). We demonstrated performance with a matrix of up to ~2.5 M pairs, and there are no evident barriers to screening at genome scale.

Each of our primary BFG-Y2H screens showed performance on par with a recent large-scale Y2H dataset resulting from two primary screens and filtering by pairwise retesting (Rolland *et al*, 2014). We also demonstrated the ability of BFG-Y2H to identify interactions for high-background "auto-activator" baits that might be considered unscreenable by current Y2H. Moreover, quantitative scoring provides greater enrichment of true interactions at higher scores, allowing user-defined trade-offs in precision versus recall. Additionally, we show by pairwise interface modeling and co-crystal structural analysis that quantitative BFG-Y2H scores are related to interaction strength. Although overall performance of BFG-Y2H rivals that of current Y2H pipelines, there are many directions for further optimization. For example, improved methods for generating barcoded strains, making strain pools with uniform representation, yeast mating and transformation, increasing the capacity of Y2H selection plates and yeast plasmid extraction efficiency, reducing PCR "jackpotting" effects, and increasing sequencing depth (Appendix Note S3). BFG-Y2H toolkit strains are compatible with other genetic screening reporters such as *ADE2* and *LacZ*. *ADE2* reporter screening reproduced 63.6% of *HIS3* reporter screening pairs (Fig EV2). Although no interaction was uniquely detected by the *ADE2* reporter assay, other reporters may in future prove complementary to *HIS3*. The Cre-mediated barcode fusion approach of BFG could, in principle, be applied to other protein interaction methods such as MAPPIT (Lemmens *et al*, 2010) or PCA (Tarassov *et al*, 2008). The "barcode carrier" destination plasmid collection described here also represents a resource reusable for other studies, either for Y2H or as a source of barcodes for other vector designs. Moreover, we described the *en masse* recombinational cloning approach as an efficient general strategy to generate barcoded reagents.

Given pools of barcoded vectors, the current BFG-Y2H pipeline requires only four days of hands-on time and seven days for the entire screen starting at the mating stage, with a single BFG-Y2H screen having the capacity to screen at least ~2.5 M ORF pairs (Appendix Fig S4). In our laboratory, one researcher can perform up to 24 parallel screens. Thus, BFG-Y2H opens the door to the exciting possibility of efficient, high-throughput screening for conditional protein interactions, for example, in the presence of alternative environments or third proteins, such as kinase or scaffold proteins (Grossmann *et al*, 2015), or in alternative genetic backgrounds.

Although the BFG technology was developed to screen protein interactions, it is applicable to other context-dependent phenotypic screens involving multiple reagents amenable to barcoding. For example, it might be applied to discover genetic interactions using high-multiplicity-of-infection lentiviral transduction to express or delete combinations of genes. There are no conceptual barriers to applying fused-barcode sequencing as a readout for multi-dimensional relationship maps in many species and cell types. Thus, BFG mapping approaches can harness the disruptive efficiencies of next-generation sequencing to generate diverse next-generation cellular maps.

## Materials and Methods

### DNA oligomers

The complete list of DNA oligonucleotides used in this study can be found in Table EV5.

### BFG-Y2H toolkit strains

The toolkit-**a** (RY1010) and toolkit-**α** (RY1030) strains were generated from the "Y-strains" Y8800 and Y8930. Y-strains were generated in C. Boone's laboratory from PJ69-4α (James *et al*, 1996) and used successfully in recent large-scale Y2H studies (Yu *et al*, 2008, 2011; Simonis *et al*, 2009; Rolland *et al*, 2014). In detail, the genotype of RY1010 is *MAT*a *leu2-3,112 trp1-901 his3-200 ura3-52 gal4Δ gal80Δ* $P_{GAL2}$-*ADE2* *LYS2::*$P_{GAL1}$-*HIS3* *MET2::*$P_{GAL7}$-*lacZ* *cyh2*$^R$ *can1Δ::*$P_{CMV}$-*rtTA-KanMX4*. The genotype of RY1030 is *MAT*α *leu2-3,112 trp1-901 his3-200 ura3-52 gal4Δ gal80Δ* $P_{GAL2}$-*ADE2* *LYS2::*$P_{GAL1}$-*HIS3* *MET2::*$P_{GAL7}$-*lacZ* *cyh2*$^R$ *can1Δ::*$T_{ADH1}$-$P_{tetO2}$-*Cre-*$T_{CYC1}$-*KanMX4*. $P_{CMV}$-*rtTA-KanMX4* was PCR-amplified from pYOGM019 (Suzuki *et al*, 2011) with O001 and O002 primers and integrated into the *CAN1* locus of the strain Y8800 to make RY1010. *Cre* ORF fragment was PCR-amplified from pSH47 (Euroscarf) with O003 and O004 primers and assembled into the BamHI-ApaI locus of pCM251 (Euroscarf) to make pNZM1010. $T_{ADH1}$-$P_{tetO2}$-*Cre-*$T_{CYC1}$ fragment was amplified from pNZM1010 with O005 and O006 primers. *KanMX4* fragment was amplified from pIS418 plasmid with O007 and O008 primers. The $T_{ADH1}$-$P_{tetO2}$-*Cre-*$T_{CYC1}$ and *KanMX4* fragments were fused by a fusion PCR with O009 and O010 primers, and the $T_{ADH1}$-$P_{tetO2}$-*Cre-*$T_{CYC1}$-*KanMX4* fragment was integrated into the *CAN1* locus of the strain Y8930 to make RY1030.

### Preparation of "barcode carrier" destination plasmid collection

For the in-yeast assembly-based BFG-Y2H, we designed the bait and prey destination plasmids to each carry two specific barcodes

(BC1 and BC2) at the PspOMI restriction site with *loxP* and *lox2272* sites (Fig EV1, Data EV1 and EV2). For each of the four barcode regions, single-stranded DNA (ssDNA) oligomer pool containing random 25-bp degenerate region flanked by universal PCR priming sites was synthesized; O011 and O012 ssDNA pools were prepared for BC1 and BC2 of bait plasmids; and O013 and O014 ssDNA pools were prepared for BC1 and BC2 of prey plasmids. The universal PCR priming sequences were designed to be unique and distinct from yeast and human chromosomes and the backbone Y2H destination vectors. Random bait-BC1, bait-BC2, prey-BC1, and prey-BC2 fragments were amplified by PCR each from ~15 × 10^18 molecules of the corresponding ssDNA with the appropriate pair of primers (O015 and O016 primers for bait-BC1, O017 and O018 for bait-BC2, O019 and O020 for prey-BC1, and O021 and O022 for prey-BC2). After the PCR, bait-BC1 and prey-BC2 fragments were flanked by *loxP* and *lox2272* sequences, and 30-bp appropriate "sealing" sequences were attached to both ends of the fragments for *in vitro* DNA assembly (Gibson *et al*, 2009). The PCR-based fragment pool preparations were designed to selectively enrich barcode sequences having high PCR efficiencies from the random ssDNA pool. The destination vectors were digested by PspOMI endonuclease and purified by a standard DNA purification. We assembled 1 μl of the bait-BC1 PCR product, 1 μl of the bait-BC2 PCR product, and 25 fmole of the linearized bait destination plasmid backbone by *in vitro* DNA assembly to make a randomly barcoded bait destination plasmid pool each having *loxP'*-BC1-*lox2272*-linker-BC2 fragment at the PspOMI locus (*loxP'* denotes the reverse complement of *loxP*). Similarly, a barcoded prey destination plasmid pool was made to have BC1-linker-*loxP'*-BC2-*lox2272* fragment.

To isolate clonal barcoded plasmids, we transformed the barcoded bait or prey destination plasmid pool into One Shot ccdB Survival 2 T1R Competent Cells (Invitrogen). The transformation samples were spread on 245 mm × 245 mm square LB–ampicillin plates and incubated overnight at 37°C so that each plate confers 500–1,500 isolated single colonies. Single colonies were picked by QPix robot (Genetix) and arrayed into 384-well cell culture plates containing 80 μl of LB–ampicillin media in each well. The cell samples were incubated overnight at 37°C in a damp box with wet paper towels to prevent sample evaporation. Glycerol stocks were made and stored in a −80°C freezer. Template DNA samples were prepared by ~20-fold dilution of the cell samples. Two RCP-PCR (Appendix Note S2) was performed to identify clonal samples with their barcode sequences and to check the integrity of *loxP* and *lox2272* sequences. The clonal samples were re-arrayed from the glycerol stocks using Biomatrix robot (S&P Robotics). Quality control of a subset of the re-arrayed samples was done by Sanger sequencing.

### In-yeast assembly-based generation of uniquely barcoded strain collection

To generate a plasmid expressing a Gal4-domain-tagged ORF and two specific barcodes, a collection of four DNA fragments were assembled: (i) a fragment containing the *ADH1* promoter and segments encoding either the DNA-binding or activation domain of Gal4; (ii) a specific ORF; (iii) a barcode locus (described further below); and (iv) a vector backbone fragment. ORF fragments were amplified from an ORFeome-level collection of bait and prey expression vectors generated in our previous study (Rolland *et al*, 2014). Each of the amplicons was generated using PCR primers that introduce 70-bp overlaps between adjacent amplicons. The DNA fragments were co-transformed into the appropriate toolkit-**a** or toolkit-**α** strain, relying on endogenous yeast DNA repair proteins to assemble the DNA fragments via overlaps between specific fragment ends. Bait ORF fragments were amplified with O023 and O024 primers, and prey ORF fragments were amplified with O025 and O026 primers. The ORF PCR products were analyzed with E-Gel 48 1% Agarose Gels (Invitrogen), and the gel band images were automatically sorted according to the expected product sizes by a Perl script developed based on ImageMagick (ImageMagick Studio) to check the quality of ORF fragments by size. We used the ORF fragments passing the Sanger sequencing and the ORF size quality control. The barcode fragments were amplified from the clonal barcode carrier destination plasmid collection (O027 and O028 primers for bait, and O029 and O030 primers for prey). The Gal4 bait and prey domain fragments were amplified by O031 and O032 primers and O033 and O034 primers, respectively. The common bait and prey backbone fragments were amplified from the destination plasmids by O035 and O036 primers and O037 and O038 primers, respectively, to generate yeast strains for the CENT space. The backbone fragments contain *E. coli* and yeast replication origins, *Amp^R* marker, and either *LEU2* (bait) or *TRP1* (prey) markers. In addition, the prey backbone fragment contains *CYH2* allowing counter-selection of the prey plasmid. For the CCC space, the backbone fragment was split into two for better PCR amplification without reduced in-yeast assembly efficiency (two fragments were produced by PCRs using O035 and O039 primer pair and O040 and O036 primer pair for bait; O037 and O041 primer pair and O042 and O038 primer pair for prey). The backbone and domain fragments were purified to adjust their molar concentrations.

Frozen-EZ Yeast Transformation II Kit (Zymo Research) was used for the library-scale in-yeast assembly reaction. In each well of 96-well plates, 10 μl of competent cells of the toolkit strain (toolkit-**α** bait and toolkit-**a** for prey), 50 fmole of the purified backbone fragment(s), 50 fmole of the purified domain fragment, 2 μl of ORF fragment PCR product, and 2 μl of barcode fragment PCR product were mixed and incubated with 100 μl of EZ3 solution for 105 min at 30 °C and 900 rpm. The plates were centrifuged at 500 *g* for 5 min, supernatants were removed, and each sample was resuspended in 100 μl of YPAD and incubated for another 105 min at 30°C and 900 rpm. The samples were then washed with sterilized water, serial dilutions of the samples were spotted on appropriate 3% agar OmniTrays (SC–Leu+Ade for bait strains and SC–Trp+Ade for prey strains), and the barcoded cells were selected for a few days at 30°C. Negative control reactions were performed at multiple random wells of each plate by replacing ORF PCR product with 2 μl of sterilized water.

### Calibration protein pair set

For BFG-Y2H screens, the 31 protein pairs were arbitrarily picked from the positive reference set (hsPRSRRSv2) of the CCSB Y2H screening pipeline with the condition that they were also included in the HI-II-14 or Lit-BM-13 sets.

## Quality control by Sanger sequencing

To check the integrity of barcodes and *lox* sites of random clones after RCP-PCR and re-arraying of clonal samples, we amplified each bait barcode region by direct colony PCR with O043 and O044 primers and prey barcode region with O045 and O046 primers. Twenty-five microliters of each PCR product was then assembled with 10 µl of an enzyme cocktail containing Antarctic phosphatase buffer (New England Biolabs), 0.5 unit of Antarctic phosphatase (New England Biolabs), and 0.5 unit of Exo I endonuclease (New England Biolabs), incubated at 37°C for 30 min, and heat-inactivated at 95°C for 5 min to remove PCR primers and phosphates of dNTP. The cleaned-up samples were sequenced by Sanger sequencing with primers O043, O044, O047, and O048 (for bait); O045, O046, O049, and O050 (for prey); and O051 and O052 (for both). To check ORFs on the entry plasmids (pDONR223), purified DNA samples were sequenced with M13 forward and reverse primers. To check ORFs on bait and prey expression plasmids, purified DNA samples were sequenced with sequencing primers O053 and O054 (for bait); O055 and O056 (for prey); and O057 and O058 (for both).

## BFG-Y2H procedure

Barcoded bait and prey strains were pre-cultured individually in 96-deep-well plates. One milliliter of SC–Leu+Ade media (1 mM excess of adenine sulfate) was used to culture each bait strain, and 1 ml of SC–Trp+Ade media was used to culture each prey strain. The deep-well plates were incubated for 2 days at 30°C and 900 rpm to saturation. The culture samples of equal volume were pooled to make bait and prey pools. The bait and prey pools were washed twice with sterilized water, resuspended in sterilized water, combined 1:1 at same $OD_{600 \text{ nm}}$ unit, and incubated for 3 h at room temperature to increase mating efficiency (Bickle *et al*, 2006). The cells were pelleted by centrifugation at 500 *g* for 4 min, and the supernatant was removed. The cell pellets were spread on YPAD plates and incubated overnight at room temperature for mating. Post-mating, cells were scraped with sterilized water, collected in a sample tube, washed twice with sterilized water, resuspended in 500 ml of SC–Leu–Trp+His+Ade media (8 mM excess of histidine and 1 mM excess of Adenine sulfate) in a 2-liter flask at 1.0 $OD_{600 \text{ nm}}$, and incubated 2 d at 30°C and 200 rpm to enrich diploid cell population until its $OD_{600 \text{ nm}}$ reaches 10. The diploid sample was washed twice with sterilized water, resuspended in sterilized water, and queried to *en masse* Y2H selection. We spread 200 µl of which $OD_{600 \text{ nm}}$ of 50-fold dilution was 1.0 (~1 × 10^8 diploid cells), each to 150-mm agar plates of SC–Leu–Trp+His+Ade (+His), SC–Leu–Trp–His+Ade (–His), and SC–Leu–Trp–His+Ade containing 1 mM amino-1,2,4-triazole (3-AT) and serial dilutions of the diploid sample on +His condition plates to confirm the sample complexity. For the larger CV and CVA screens, 20 150-mm plates were used for each selection to ensure sufficient coverage of the complexity. The selection plates were incubated for 2–3 days at 30°C. For each condition, selected cell samples were scraped with sterilized water, collected in a sample tube, washed twice with sterilized water, resuspended in 5 ml of SC–Leu–Trp+His+Ade containing 10 µg ml$^{-1}$ doxycycline at 1.0 $OD_{600 \text{ nm}}$, and incubated overnight at 30°C until its $OD_{600 \text{ nm}}$ exceeds 5.0 to allow the *in vivo* Cre-mediated barcode fusion within each cell. Plasmid DNA was extracted from yeast cells

using Charge Switch Plasmid Yeast Mini Kit (Invitrogen). The BC1-BC1 and BC2-BC2 fused barcodes were amplified by Phusion High-Fidelity PCR Master Mix with HF Buffer (New England Biosciences) and appropriate sets of primers containing Illumina paired-end sequencing adapters with index tags for multiplexed sequencing (Table EV5) with the PCR protocol: (i) 98°C for 30 s; (ii) 24 cycles of 98°C for 10 s, 60°C for 10 s, and 72°C for 1 min; (iii) 72 °C for 5 min; and (iv) 4°C forever. The PCR volume and template DNA amount were designed to ensure the sample complexity for each screen (one 40 µl reaction with 2 ng template DNA to amplify each of BC1-BC1 and BC2-BC2 fused barcodes for CENT and CCC screens and 20 40-µl reaction each with 2 ng template DNA for CV and CVA screens, Appendix Note S3). The PCR products were size-selected by E-Gel SizeSelect 2% Agarose gel (Invitrogen) and quantified by qPCR for Illumina sequencing. The reads were demultiplexed and fused barcodes were counted by aligning the reads to the primer sequences and DNA barcodes using BLAST+ program with the short-read option and E-value cutoff of 1e–10.

## Pairwise retesting of candidate interactions

Pairwise Y2H retesting was performed based on the previously established protocol (Rolland *et al*, 2014). Bait and prey expression plasmids were prepared by Gateway LR cloning. Each protein pair was tested in the toolkit strain background and the Y-strain background in quadruplicate. For each reaction, bait cells and prey cells were combined 1:1 in PCR plate wells, in 120 µl of YPAD, and incubated overnight at 30°C and 200 rpm. Diploid cells were enriched by inoculating 10 µl of each mated sample to 120 µl of SC–Leu–Trp+His+Ade in 96-well cell culture plates and incubating overnight at 30°C and 200 rpm. Using a Biomatrix robot (S&P Robotics), we spotted the 4 replicate samples on SC–Leu–Trp–His+Ade+3-AT agar plates (384 spots per plate) and SC–Leu–His+Ade agar plates containing 1 mg l$^{-1}$ cycloheximide (CHX) to identify auto-activators. Additionally, bait cells were mated with opposite-mating-type cells harboring "null ORF" prey plasmids.

## *Gaussia princeps* luciferase protein complementation assays

*Gaussia princeps* luciferase protein complementation assay (GPCA)-based validation assay was performed to validate the in-yeast assembly-based BFG-Y2H hits as described previously (Remy & Michnick, 2006; Cassonnet *et al*, 2011). In order to define the threshold of GPCA signal for each validation screen, positive controls were obtained from the overlap of the GPCA-tested space with the union of the systematic Y2H human interactome HI-II-14 dataset (Rolland *et al*, 2014) and the high-quality literature-curated protein interaction dataset Lit-BM-13 (Rolland *et al*, 2014). Fifty-four and 36 arbitrarily protein pairs that scored poorly in CENT and CCC screens, respectively, were selected as negative controls (Table EV2). The luciferase signal threshold for each validation assay was defined by the best balance of recall and precision according to the MCC.

## *En masse* recombinational cloning-based creation of barcoded strain pools

To scale up the BFG-Y2H method, we generated pools of barcode carrier destination plasmids encoding random barcodes and *lox* sites

at the downstream of ORF region (SacI restriction site). Pools of random barcode fragments were prepared by the protocol described above (primers listed in Table EV5) and assembled with SacI-digested bait or prey destination plasmid DNAs by *in vitro* DNA assembly. The reaction product was used to transform One Shot ccdB Survival 2 T1R Competent Cells (Invitrogen), and the bacterial colonies were pooled to recover randomly barcoded bait or prey destination plasmids. Bacterial strains harboring Gateway entry plasmids for a given ORF space were arrayed on LB–spectinomycin plates using a BioMatrix robot (S&P Robotics), incubated overnight at 37°C, pooled, and harvested for plasmid DNA purification. One hundred and fifty nanograms of bait or prey barcode carrier destination plasmid pool and 150 ng of entry ORF pool dissolved in TE buffer were combined with 1 μl of Gateway LR Clonase II (Life Technologies) in a total of 5 μl reaction volume and incubated overnight at room temperature. In each of the following 4 days, 150 ng of destination plasmid pool in 4 μl of TE and 1 μl of the enzyme was added to each reaction and incubated overnight at room temperature to saturate the reaction for each ORF and normalize ORF-dependent Gateway LR reaction efficiencies. The randomly barcoded bait or prey expression plasmid pool was transformed to NEB 5-alpha Competent *E. coli* cells (New England Biolabs), and colonies were isolated and arrayed in 384-well format on LB+ampicillin plates using a BioMatrix robot (S&P Robotics). Barcode sequences and ORFs of the entire colonies were then identified by kiloSEQ (seqWell Inc.). According to the sequencing result, barcoded clones were selected, re-arrayed, and incubated overnight on LB+ampicillin plates to reduce the bias in number of different barcodes per each ORF. An expression plasmid pool of known barcode and ORF combinations was then purified from a pool of the bacterial cells. BFG-Y2H-ready bait or prey strain pool was generated by a single-yeast transformation reaction by combining 10 μg of barcoded bait or prey expression plasmid pool with 500 μl of RY1030 or 1010 competent cells (increasing the reaction scale of Frozen-EZ Yeast Transformation II Kit, Zymo Research). After 90-min outgrowth with 5 ml of YPAD, cells were resuspended in 300 μl of sterilized water, spread on two 150-mm selective plates (150 μl each; SC–Leu+Ade for bait and SC–Trp+Ade for prey), and incubated for 2–3 days at 30°C. Selected colonies were pooled and subjected to the BFG-Y2H procedure.

### Reported protein interaction dataset

Reported Y2H-positive human protein interaction dataset HI-II-14 and the Lit-BM-13 dataset were downloaded from the CCSB Human Interactome Database (http://interactome.dfci.harvard.edu), and reported interactions by orthogonal assays were obtained from the BioGRID database (Chatr-Aryamontri *et al*, 2013) Release 3.1.91.

### Statistical analysis

Sample correlations were evaluated using Pearson's correlation coefficient. Sample enrichments were evaluated by Fisher's exact test or Pearson's chi-squared test. Comparison of distributions of GPCA or interaction scores was performed using the Mann–Whitney *U*-test.

### Plasmids

Plasmids used in this study can be found in Data EV1 and EV2.

### Data availability

The protein interactions from this publication have been submitted to the IMEx (http://imexconsortium.org) consortium through IntAct (Orchard *et al*, 2014) and assigned the identifier IM-25015. The data used for the method evaluation is found under the IMEx identifier IM-23318. Raw read counts can be found in Data EV3.

**Expanded View** for this article is available online.

### Acknowledgements

This work was supported by the Krembil Foundation (F.P.R. and L.P.), the Avon Foundation (F.P.R.), the Ontario Research Fund (F.P.R. and L.P.), the Canada Excellence Research Chairs Program (F.P.R.), and by US National Institutes of Health (NIH) grants HG001715 (M.Vidal, D.E.H and F.P.R.) and HG004233 (a "Centers of Excellence in Genomic Science" award; M.Vidal and F.P.R.). N.Y. was supported by a JSPS Fellowship (Research Abroad), Japan Society for the Promotion of Science, a Banting Postdoctoral Fellowship, National Sciences and Engineering Research Council of Canada, Japan Science and Technology Agency (JST) PRESTO grant, JSPS grant 15K18466, Astellas Foundation for Research on Metabolic Disorders, Shimazu Foundation, Life Science Foundation of Japan, and by Nestlé Nutrition Council, Japan. L.P. was supported by Canadian Institutes of Health Research (CIHR) grants MOP-123468 and MOP-130507. I.S. was supported by grants from the CQDM/OCE Explore Program, Ontario Genomics Institute, Canadian Cystic Fibrosis Foundation, Canadian Cancer Society, Pancreatic Cancer Canada, University Health Network and Genome Canada DIG1 and DIG2 programs. Sequencing was performed at the Donnelly Sequencing Centre, University of Toronto. We are grateful for early discussions with J. Weissman, for Y-strains from C. Boone (University of Toronto), and for advice and assistance from B. Andrews, M. Chee, G. Church, A.-C. Gingras, L. Heisler, J. Liu, N. Mohammad, L. Ng, C. Nislow, A. Rosebrock, K. Salehi-Ashtiani, B. Steen, and D. Torti; members of the Roth Lab, especially C. Cheung, A. Karkhanina, S. Sun, Y. Suzuki, M. Tasan and J. Weile; members of the DFCI Center for Cancer Systems Biology (CCSB); and members of the Pelletier laboratory, especially M. Bashkurov, D. Comartin, J. Goncalves, S. Lawo, B. Mojarad, and C. Yeh.

### Author contributions

NY, JCM, SBO, and FPR developed the BFG-Y2H technology; NY, EP, and FPR prepared the manuscript; NY, EP, and FPR designed the experiments; NY, EP, MVe, TT, and MG made the barcode carrier plasmid collection; NY, EP, MVe, and DS performed the BFG-Y2H screens; NY, EP, AY, MVe, SL, JFR, CW, and JJK performed pairwise retesting; AGC performed sequencing; RM and PA carried out computational structural analysis; NY, EP, and JW performed other computational analyses; YJ performed GPCA analysis; MK, NS, SY, LM, JS, Y-CL, HY, PB, IS, TH, MAC, LP, DEH, and MVi provided reagents, technical support, and advice.

### Conflict of interest

The authors declare that they have no conflict of interest.

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
