## [Review Process File · Molecular Systems Biology]

Pooled-matrix protein interaction screens using Barcode Fusion Genetics

Nozomu Yachie, Evangelia Petsalaki, Joseph C. Mello, Jochen Weile, Yves Jacob, Marta Verby, Sedide B. Ozturk, Siyang Li, Atina G. Cote, Roberto Mosca, Jennifer J. Knapp, Minjeong Ko, Analyn Yu, Marinella Gebbia, Nidhi Sahni, Song Yi, Tanya Tyagi, Dayag Sheykhkarimli, Jonathan F. Roth, Cassandra Wong, Louai Musa, Jamie Snider, Yi-Chun Liu, Haiyuan Yu, Pascal Braun, Igor Stagljar, Tong Hao, Michael A. Calderwood, Laurence Pelletier, Patrick Aloy, David E. Hill, Marc Vidal and Frederick P. Roth

Corresponding authors: Nozomu Yachie, University of Tokyo and Frederick P. Roth, University of Toronto

Review timeline:

Submission date:	27 October 2015
Editorial Decision:	07 December 2015
Revision received:	13 February 2016
Editorial Decision:	09 March 2016
Revision received:	11 March 2016
Accepted:	18 March 2016

Editor: Maria Polychronidou

Transaction Report:

1st Editorial Decision

07 December 2015

Thank you again for submitting your work to Molecular Systems Biology. We have now heard back from the three referees who agreed to evaluate your manuscript. Overall, the referees acknowledge that the presented approach seems interesting. However, they raise a series of concerns, which should be carefully addressed in a revision of the manuscript. The reviewers' recommendations are rather clear so there is no need to repeat all the issues listed below.

As you will see, Reviewer #2 has a series of concerns regarding the experimental design and the quality of the proposed method. We have circulated the reports to all reviewers as part of our 'pre-precision cross-commenting' policy. During this process, Reviewer #1, mentioned that "while no newly developed method is expected to be perfect, it is very important to provide a "pros and cons" table that mentions scale, time, cost, retest rate, false negatives etc. and that includes this method as well as the Vidal method and other methods" (as also suggested in his/her review). S/he also mentioned that the major issues raised by Reviewer #2 could be addressed within the scope of a major revision. During cross-commenting, reviewer #3 agreed with both Reviewers #1 and #2 that the manuscript should be carefully edited in order to be better accessible to a general audience and to ensure that important experimental details and data analyses are clearly described. S/he also provided a suggestion for addressing the concern regarding fitness raised by Reviewer #2, pasted here for your reference: "In my opinion, as an important control for the authors to address the fitness

concern, they should take at least three pairs of proteins of known high, medium, and low affinity values and test their growth rate after mating in a pool."

REFeree COMMENTS

Reviewer #1:

This is a very nice paper describing a novel and elegant Y2H method in which bait and prey-derived sequences are combined together into a single DNA molecule that can be read by sequencing. It promises to further increase the throughput of the Y2H method, which is needed to contemplate completion of a human (or model organism) reference interactome.

Specific comments:

There are more Y2H strategies that could be mentioned, including smart pooling (PMID: 19447967, 17589517), and screening in a pair-wise, yet high-throughput manner (PMID: 23791784; 19521828). In light of this, it would be useful to have a pros and cons table for the different methods, including BFG-Y2H.

The paper is densely written and sometimes hard to follow with tons of names/wording, abbreviations and acronyms. For instance, the authors dive into the details of their method in the first section on page 3, without providing a conceptual overview/rationale. I strongly suggest they first have a conceptual framework, with no technical details, and then a technical overview with the detailed naming of all the players (uptag/downtag - lox this lox that, etc is hard to read when the reader just gets ready to learn about this new approach). The conceptual overview should be accompanied by a figure just showing the recombination events, starting and end molecules (conceptually), as in Figure 1, but larger and without the yeast contour.

Figure 2 is not clearly described in the main text, while the authors include volumes of reagents used (not necessary in figure), essential elements for growth and generation of the correct bait/prey are not indicated. Also it is not clear how figure 2b is showing that "A plasmid-encoded selectable marker allowed enrichment for cells bearing correctly assembled BFG- Y2H plasmids (Figure 2b)".

Does using a ranking cutoff lead to a greater likelihood of missing weaker (but potentially valid) interactions? Some (weak) evidence for a correlation with interaction strength is presented in Figure 4e. It would be useful if the authors could comment on how they envision to capture such interactions.

Finally, I am sorry to say that it was torture to read the densely packed, small font text, not to mention to look at the tiny figures in an ocean of white space!

Reviewer #2:

Yeast two hybrid screens (Y2H) have been used for the last 35 years to discover proteins that interact. These work by mating bait and a prey strains together and screening these combinations via reconstruction of an active transcription factor and growth on plates. For extremely large interaction sets such as the human interactome, the throughput of pairwise screening falls short by one or two orders of magnitude. To overcome this problem, this paper develops and tests a barcode-based method to construct and screen large Y2H libraries in pools. On each Y2H bait or prey plasmid, a unique identifier surrounds a pair of non-interacting loxP sites. When two plasmids are incorporated into a cell, Cre is induced and some fraction of plasmids swap the DNA that is sandwiched between the two loxP sites. By sequencing the DNA surrounding the loxP sites, the specific pair of plasmids in each cell can be discovered. The authors first test this system on a matrix of ~25K protein pairs, the huCENT matrix. They report quite disappointing results including a surprising and unexplained nearly uniform distribution in barcode counts and a sequencing run where the vast majority of sequencing reads map to autoactivators. Instead of addressing these problems directly by changing the experimental protocol, the authors decide to repeat the screen with the seven strongest

autoactivators removed. On this rerun, other unremoved autoactivators appear to constitute the vast majority of reads, however a fraction of putative interactions also appear to have sufficient reads. Read counts and their distributions are never reported, and without this information, cost savings of this sequencing-based method is impossible to judge. To identify true positives, the authors devise a complex scoring statistic that uses the fifth highest ranking interaction signal for each protein pair in one condition across numerous replicates of the the experiment. How well this statistic (which is tuned for the huCENT matrix) performs on other matrices, or if a different ad hoc method is employed, is never discussed. The authors then perform the same screen on a larger huCCC interaction matrix and the top 100 interactions from these matrices were retested to test the assay performance, resulting in ~50% positive retest scores. These 100 were also retested using a split luciferase protein fragment complementation system, finding approximately the same 50% overlap. How a traditional Y2H screen performs in this luciferase retest is never discussed. The authors then scaled up to larger interaction matrices and compare these results to traditional Y2H, finding that the barcode-based method performed worse, in most cases identifying only ~50% of the interactions of traditional methods (with an expected lower false positive rate).

Overall, I found this manuscript to be a frustrating read that lacks or never clearly articulates the information that is necessary to make a critical judgment of the method. One addressable problem is that the manuscript is filled with an enormous number of acronyms, many of which are unnecessary, and many others of which might be appropriate in specialist journals, but not for a general audience. I often found myself hunting all over the manuscript to decipher a single sentence that contains 5 or 6 acronyms, or a figure with acronyms not described in the legend. However, there appear to be much deeper problems, particularly in the experimental design. The authors plate their double plasmid pools at a low density and grow these for 2-3 days on selective plates, meaning any design with a high fitness will have a long time (an important but uncharacterized number of generations) to expand exponentially within the population. Thus, by the time the pool is sequenced, the vast majority of cells are a small sliver of the most fit strains. As a practical matter, these fittest strains will constitute the vast majority of sequencing reads, meaning that in order to detect the lower fitness interactions by sequencing, either an enormous amount of sequencing must be "wasted" on resequencing of the fittest strains, or a new pool must be constructed with these high fitness strains thrown out. Making matters worse, it appears as though the vast majority of the highest fitness strains are autoactivators, meaning that most of the sequencing costs will go to experimental noise. With no clear way to identify autoactivators a priori, it is unclear how this problem can be overcome, and it was undiscussed in the larger matrix screens. Even with autoactivators removed, this assay will always be biased towards characterizing the strongest protein interactions. This effect provides a potential explanation for its poor performance when compared to traditional Y2H and why retesting of 100 interactions resulted in only 50% retest rate (competition between only the fittest 100 strains will cause lowest fitness strains in this group to decline in frequency). Much of the rest of this manuscript is statistical methods built around this flawed experimental design in order to improve the predictive value. However, if the aim of this method is to expand throughput to characterize the full human interactome, this simply won't work because lower fitness interactions will always be missing.

Additional major criticisms

1) Given that the ultimate readout of this assay is sequencing count for each fusion construct, there is surprisingly little discussion of the methods and caveats around deep sequencing of fused plasmids. Missing is the PCR methods (cycles, etc.), number of reads for each experiment, and the analysis pipeline. How much of a problem is PCR jackpotting? Are sequencing errors ignored or clustered to a known barcode? How much sequencing error is tolerated? Does count error derived from sequencing resemble Poisson sampling? Does the size of the fusion impact what gets PCR amplified? These questions speak directly to the scalability of the system.

2) Figure 3a shows a uniform distribution of fused barcode copy numbers. This is a crucial distribution that speaks to how many interactions one would never have a chance to see at various library sizes. One would naively expect that this should be roughly normal in His⁺, as each fusion barcode should have an ~ equal chance of making it in the pool without any selection. A uniform distribution is unexpected and never explained. Why is 0 excluded from these plots? Zero is the most important data point to interpreting how well the system works.

3) There needs to be a better characterization of how all of the steps of making the double plasmid libraries alters barcode frequencies. How many generations in each step? How does noise introduced during this process impact the screen?

4) The authors need to address more quantitatively the limitations of their screen. Crucially, how many generations pass during competitive growth in -His? What is the fitness difference on -His between a strain with a strong interaction vs. one with no interaction. Based on these estimates, in a screen with only these two strains, what fraction of the population would one expect to be the strong interaction strain at the time of sequencing? Can simulations be used to understand which lower-fitness interactions would not be detected?

5) Figures and figure legends are often incomplete.

6) I would expect that plasmid size has an impact on the large matrix assays. For example large plasmids might be underrepresented in cell pools because it is more difficult to insert them into cells. Plasmid copy number variation may also be in issue. This could mean that many PPIs have no chance of being detected. However, these caveats are never discussed or addressed.

7) "Accurately estimating the effects of Y2H selection on each X-Y pair must account for several factors: 1) uneven strain abundance in the initial DB-X and AD-Y libraries; 2) potential growth effects of DB-X and AD-Y expression; and 3) slight barcode-dependent differences in amplification and sequencing." In addition random growth processes, PCR jackpotting, and sampling from the sequencer are important and should be discussed.

Minor criticisms

IS stands for "interaction score", yet IS score (interaction score score) is used throughout the paper and figures

Page 10 - "it might be applied to discover genetic interactions using high-multiplicity-of-infection lentiviral transduction to express or delete combinations of genes." Untrue in its current form. Genetic interactions require a fitness measurement, but this assay has a digital readout of whether each strain is in the pool or not.

Figure 3a - 10^{-2} appears twice on the x axis

Figure 3b, c - The blue color scale makes it extremely difficult to identify positives, or see differences

Figure 3d - axis labels of the number of reads are critical to interpreting these plots

Figure 3f - insufficient legend

Figure 5c - A histogram is more straightforward and interpretable than a cumulative % plot

Figure EV2b - RY and Y or unexplained

Figure EV2d - I'm confused. Are strains each containing a bait and a prey mated?

Appendix S4e- Sequencing costs of the library are missing

Reviewer #3:

The manuscript, entitled "Pooled-matrix protein interaction screens using Barcode Fusion Genetics" by Yachie et al, describe a new yeast two-hybrid method that enables multiplexed, one pot screen for protein-protein interactions. In their Barcode Fusion Genetics-Yeast Two-Hybrid (BFG-Y2H), the bait library (i.e., DB-X) is prepared in a haploid yeast strain in which each strain carries a

plasmid that expresses a DB-X protein upon doxy induction and also contains UPTAG DNA barcode flanked by direction LoxP and Lox2272 recombination sites. Similarly, the prey library (i.e., AD-Y) is prepared in a haploid yeast strain in which each strain carries a plasmid that expresses an AD-Y protein upon doxy induction and also contains DNTAG DNA barcode flanked by direction LoxP and Lox2272 recombination sites. To screen for protein-protein interactions, the strains of these two libraries are pooled and mated en mass to create diploid cells, representing all X-Y combinations between the bait and prey proteins. When a pair of protein interacts in the diploid cells, the strain becomes His⁺ and survives on the His dropout media. By turning on Cre, recombination happens among LoxP and Lox2272 sites, generating an "Up-Up" barcode on the AD-Y plasmid and a "Dn-Dn" barcode on the DB-X plasmid. Using deep-seq, the authors could readily identify the pairs of positive PPI relationships. To demonstrate its application, the authors first tested a set of human centrosomal and centrosome-associated proteins, comprised of 143 DB-X and 162 AD-Y. Using non-selective media, they estimated that the screen was close to saturation and the sampling sensitivity was high. To score protein-protein interactions on the selective media, they normalized the sequencing read counts to obtain a quantitative interaction score (IS). They concluded that protein pairs with high IS scores are enriched for known interactions. Using another set of protein constructs implicated in cancer, the authors evaluated the performance of this BFG-Y2H approach in detecting auto-activators, those that do not require the prey to be able to activate His gene. They concluded that this new approach was sensitive enough to detect homodimer formation. They then went on to demonstrate a more efficient en mass strategy for producing barcoded BFG-Y2H constructs and compared it with the traditional Y2H approach.

In my opinion, this new design of pooled Y2H screen is ingenious because it allows simultaneous screens for any combinations of potential protein-protein interactions. In the past ten years, Vidal and colleagues have improved the state-of-art Y2H approach and applied it to map out global protein-protein interactions that laid the foundation of biological networks for many research fields. However, it is still quite labor-intensive and time-consuming. For this very reason, the human protein-protein interaction networks are still far from completion. This new BFG-Y2H approach now holds the potential to survey the entire 20,000 by 20,000 combinations of protein-protein interactions in humans, as well as in other important model organism. I envision it will be a tremendous boost to the entire biological community. I was also quite impressed by their careful design and test of this new approach. The comparison with the state-of-art Y2H approach serves as a solid validation for this new methodology. One point the author raised in the manuscript was very intriguing to me; that is the IS scores could indicate the affinity of a protein-protein interaction as it represents the normalized deep-sequencing read counts. I wonder whether the authors have taken protein pairs of either high versus low IS scores and measure their true binding affinity using standard methods, such as SPR and OCTET? In my opinion, if it holds true, it would be very valuable information to have for any pairs of protein-protein interactions. Other than this point, I don't have any more issues. This work suits very well with the readership of MSB.

1st Revision - authors' response

13 February 2016

Editorial comments:

Thank you again for submitting your work to Molecular Systems Biology. We have now heard back from the three referees who agreed to evaluate your manuscript. Overall, the referees acknowledge that the presented approach seems interesting. However, they raise a series of concerns, which should be carefully addressed in a revision of the manuscript. The reviewers' recommendations are rather clear so there is no need to repeat all the issues listed below.

As you will see, Reviewer #2 has a series of concerns regarding the experimental design and the quality of the proposed method. We have circulated the reports to all reviewers as part of our 'pre-decision cross-commenting' policy. During this process, Reviewer #1, mentioned that "while no newly developed method is expected to be perfect, it is very important to provide a "pros and cons" table that mentions scale, time, cost, retest rate, false negatives etc. and that includes this method as well as the Vidal method and other methods" (as also suggested in his/her review). S/he also mentioned that the major issues raised by Reviewer #2 could be addressed within the scope of a major revision. During cross-commenting, reviewer #3 agreed with both Reviewers #1 and #2 that the manuscript should be carefully edited in order to be better accessible to a general audience and

to ensure that important experimental details and data analyses are clearly described. S/he also provided a suggestion for addressing the concern regarding fitness raised by Reviewer #2, pasted here for your reference: "In my opinion, as an important control for the authors to address the fitness concern, they should take at least three pairs of proteins of known high, medium, and low affinity values and test their growth rate after mating in a pool."

We thank the editor and reviewers for a thorough and thoughtful review process. The manuscript has been revised extensively to address the reviewer comments. Below we include appoint-by-point response, indicating changes made to the revised manuscript.

Reviewer #1:

This is a very nice paper describing a novel and elegant Y2H method in which bait and prey-derived sequences are combined together into a single DNA molecule that can be read by sequencing. It promises to further increase the throughput of the Y2H method, which is needed to contemplate completion of a human (or model organism) reference interactome.

We appreciate the positive feedback.

Specific comments:

Reviewer Comment 1.1

There are more Y2H strategies that could be mentioned, including smart pooling (PMID: 19447967, 17589517), and screening in a pair-wise, yet high-throughput manner (PMID: 23791784; 19521828). In light of this, it would be useful to have a pros and cons table for the different methods, including BFG-Y2H.

Response 1.1

According to the reviewer's suggestion, we have added Appendix Table S4 comparing recent high-capacity Y2H methods. Although we restricted this table to Y2H methods for measuring protein-protein interactions (and therefore did not include the suggested references related to the yeast one-hybrid method for detecting DNA-protein interactions), we would be happy to expand the table to include methods for detecting different kinds of interactions at the Editor's discretion.

Reviewer Comment 1.2

The paper is densely written and sometimes hard to follow with tons of names/wording, abbreviations and acronyms. For instance, the authors dive into the details of their method in the first section on page 3, without providing a conceptual overview/rationale. I strongly suggest they first have a conceptual framework, with no technical details, and then a technical overview with the detailed naming of all the players (uptag/downtag - lox this lox that, etc is hard to read when the reader just gets ready to learn about this new approach). The conceptual overview should be accompanied by a figure just showing the recombination events, starting and end molecules (conceptually), as in Figure 1, but larger and without the yeast contour.

Response 1.2

With hindsight, we agree and thank the reviewer for these constructive comments. We have now included a paragraph with a more accessible description of the method and a figure (Figure 1) showing the conceptual framework. We have also substantially reduced abbreviations/acronyms, and added clarifying comments throughout the text to make it more accessible to a wider audience.

Reviewer Comment 1.3

Figure 2 is not clearly described in the main text, while the authors include volumes of reagents used (not necessary in figure), essential elements for growth and generation of the correct bait/prey are not indicated. Also it is not clear how figure 2b is showing that "A plasmid-encoded selectable marker allowed enrichment for cells bearing correctly assembled BFG- Y2H plasmids (Figure 2b)".

Response 1.3

We have expanded our explanation of Figure 2 (now Figure 3) in the main text and legend, and moved some experimental details to supplementary material. Figure 3b shows that yeast cells transformed with all four of the DNA plasmid fragments required for proper in-yeast assembly grow in selective media and that negative control transformations without the ORF-containing fragment (highlighted in yellow) do not.

Reviewer Comment 1.4

Does using a ranking cutoff lead to a greater likelihood of missing weaker (but potentially valid) interactions? Some (weak) evidence for a correlation with interaction strength is presented in Figure 4e. It would be useful if the authors could comment on how they envision to capture such interactions.

Response 1.4

As with all other methods for detecting protein interaction, sensitivity is diminished for weaker interactions. We have tried to better emphasize our main point that, unlike interaction scoring methods without quantitative scoring, the quantitative BFG-Y2H score allows the user to adjust the threshold to increase the coverage of weaker interactions. Because lowering the stringency to include weaker interactions will also admit more false positives, the appropriate threshold will depend on the tolerance of the user for false positives, and their willingness to subject candidate interactions to individual retesting.

Reviewer Comment 1.5

Finally, I am sorry to say that it was torture to read the densely packed, small font text, not to mention to look at the tiny figures in an ocean of white space!

Response 1.5

We apologize for the torture! We have now expanded the images where possible, and increased the font size and line spacing in the main text (12 pt). We have also increased the font size in the figures – except for a few figures in the Appendix, these should all now be at least 11-12 pt when a figure is printed on letter-sized paper with a landscape orientation. We are happy to work with the Editor to rapidly make any further figure improvements.

Reviewer #2:

Reviewer Comment 2.1

Yeast two hybrid screens (Y2H) have been used for the last 35 years to discover proteins that interact. These work by mating bait and a prey strains together and screening these combinations via reconstruction of an active transcription factor and growth on plates. For extremely large interaction sets such as the human interactome, the throughput of pairwise screening falls short by one or two orders of magnitude. To overcome this problem, this paper develops and tests a barcode-based method to construct and screen large Y2H libraries in pools. On each Y2H bait or prey plasmid, a unique identifier surrounds a pair of non-interacting loxP sites. When two plasmids are incorporated into a cell, Cre is induced and some fraction of plasmids swap the DNA that is sandwiched between the two loxP sites. By sequencing the DNA surrounding the loxP sites, the specific pair of plasmids in each cell can be discovered. The authors first test this system on a matrix of ~25K protein pairs, the huCENT matrix. They report quite disappointing results including a surprising and unexplained nearly uniform distribution in barcode counts...

Response 2.1

We regret that this point was not better explained. In the original manuscript we did indeed say that “The distribution of used barcode counts from non-selective media was nearly uniform across all X-Y pairs.” However, this was a poor choice of words, made more confusing by our use of a log-scaled y-axis in Figure 3a (now Figure 4a). The point we intended to make was that we could, even with low-saturation sequencing, detect fused-barcode pairs for a large fraction of bait/prey combinations. That the complexity of the starting population was high was good news, suggesting that most bait/prey pairs were sufficiently represented in the pre-selection population. We have now clarified the wording. We have also changed the y-axis scale in Figures 4a and Figure EV4d from log- to linear-scale, and added a distribution with the inferred abundance of bait/prey combinations, which is expected to be more accurate than direct observation from low. The revised text and Figures 4a and EV4d now more clearly show that the distribution of fused-barcode abundance has a characteristic peak, and that we can infer that the vast majority of bait/prey pairs were well-represented in the unselected population.

Reviewer Comment 2.2

...and a sequencing run where the vast majority of sequencing reads map to autoactivators. Instead of addressing these problems directly by changing the experimental protocol, the authors decide to

repeat the screen with the seven strongest autoactivators removed. On this rerun, other unremoved autoactivators appear to constitute the vast majority of reads, however a fraction of putative interactions also appear to have sufficient reads.

Response 2.2

We appreciate the reviewer's comments, and have modified the manuscript to make the following points more clearly.

In current state-of-art Y2H technologies, baits with a high background level of *HIS3* activity ('auto-activators') can be removed *a priori* by testing all of the baits with a 'null' prey (Dreze et al, *Methods in Enzymology* 2010). In BFG-Y2H, all of the query ORF's can be pooled in a single reaction without the prior removal of auto-activators. No significant difference in best performance recapturing the 'Union' interactome dataset (MCCmax) was observed between screens with and without the seven strongest auto-activators (Appendix Figure S2). Furthermore, we could systematically quantify background auto-activity levels for all of the bait proteins (appendix Note S4). By normalizing signals according to the auto-activity levels, interactions were scored even for protein pairs where baits were identified as auto-activators by individual Y2H testing.

For protein pairs that were within the top 100 in CENT screen and which also scored as auto-activators by individual Y2H testing were subjected to the orthogonal *Gaussia princeps* luciferase complementation assay (GPCA). These yielded significantly higher GPCA signal intensities than protein pairs that scored negatively in BFG-Y2H (Figure 5d).

It is true that barcode counts from these high-background baits compose a disproportionate share of each sequencing run, so that additional reads may be required to ensure sufficient coverage of lower-background baits. Like genome sequencing, where there is a higher cost associated with 'finishing' a genome, there is a higher cost associated with completing an interaction screen for the most problematic baits. However, the analogous cost for covering the high-background baits with BFG-Y2H is not prohibitive. By removing autoactivators in a 2kx 2k matrix screen, we might save the cost of a sequencing run (~\$1000, which is ~25% of the overall cost of performing a screen), but we would also lose the opportunity to test those baits that are high-background (which conservatively represent 5-10% of all baits). Additionally, in order to identify the auto-activator baits we would have to initially perform either a pairwise test, which is laborious, or a screen of all baits with AD-NULL, which would add back some of the sequencing cost that we might have saved by removing high-background baits.

Although there may be opportunities to optimize our treatment of high-background baits, it is worth stepping back to look at the big picture: Even with auto-activator baits included, we show that BFG-Y2H identifies interactions with quality on par with the latest state-of-art Y2H technology (Figure 7), at <30% of the cost of state-of-the-art Y2H.

Reviewer Comment 2.3

Read counts and their distributions are never reported, and without this information, cost savings of this sequencing-based method is impossible to judge.

Response 2.3

The revised manuscript now shows read count distributions in Figure EV4a and b while Fig 4a and EV4d provide frequencies along with normalized barcode counts so that the distribution of different selective conditions can be compared. Row read counts of fused-barcodes are now also provided in Appendix Data S3. Please note that, for the purpose of scoring, the relative abundance of each diploid strain in the non-selective (+His) condition was modeled as the simple product of the relative abundances of the parental haploid strains using the row- and column-totals from fused barcode counts (Appendix Note S4 and Figure EV4c). This avoids the requirement that each particular barcode combination in the non-selective condition be well represented with expensive high-saturation sequencing of fused-barcodes for the non-selective library.

Despite low-saturation sequencing in the non-selective condition, amongst all of the ORF pairs in CENT screen, 99.8% of the ORF pairs were directly observed by at least one fused-barcode tag in the non-selective (+His) condition across two screening variants (Figure EV4e). The Monte-Carlo

simulation of the BFG-Y2H pipeline also supported the read depths of the screens (Appendix Note S3 and Appendix Figure S1). We have now also provided the raw read counts as Appendix Data S3.

Reviewer Comment 2.4

To identify true positives, the authors devise a complex scoring statistic that uses the fifth highest ranking interaction signal for each protein pair in one condition across numerous replicates of the the experiment. How well this statistic (which is tuned for the huCENT matrix) performs on other matrices, or if a different ad hoc method is employed, is never discussed.

Response 2.4

We apologize for not having clarified this in the text. The CENT screening was performed twice (with and without the seven strong auto-activators). Each protein pair had multiple replicates at both barcode-pair and strain-pair levels. We have now clarified these points and the idea behind the statistics (Appendix Note S4). We separately optimized the parameters to achieve the best performance of the scoring method for each given screen against Y2H interactions reported in other high-throughput screens. Note that optimization of scoring in this way does not make use of the literature-derived interactions that we later use as an independent measure of performance. We have clarified these points in the manuscript.

While use of the *n*th-ranked value among replicates may seem *ad hoc*, this is mathematically equivalent to estimating a value by picking a percentile value. Although use of the 50% value (median) is common, there is ample precedent for using other percentile thresholds for estimation from a distribution, e.g., for estimating expression level from a collection of oligonucleotide ‘features’ on an Affymetrix array. For Affymetrix arrays, the rationale was that a subset of oligos yield no signal and are uninformative, so that examining values at higher percentiles can be a proxy for taking the median of the subset of high-signal informative features.

Reviewer Comment 2.5

The authors then perform the same screen on a larger huCCC interaction matrix and the top 100 interactions from these matrices were retested to test the assay performance, resulting in ~50% positive retest scores.

Response 2.5

This is correct. We note that pairwise Y2H retest success rate of ~50% are in accordance to that observed with the state-of-the-art Y2H pipeline (Rual et al, *Nature* 2005; Rolland et al, *Cell* 2014). Moreover, results in Fig 5c (formerly Fig EV4c) suggest that the 50% retest rate we observed is an underestimate of the fraction of BFG-Y2H hits that are *bona fide* interactions.

Reviewer Comment 2.6

These 100 were also retested using a split luciferase protein fragment complementation system, finding approximately the same 50% overlap. How a traditional Y2H screen performs in this luciferase retest is never discussed.

Response 2.6

The validation rate of our reported hits by GPCA was 74% for the CENT screen and 46% for the CCC screens with only 4% and 3% of tested BFG-negative pairs appearing as positives in the luciferase assay.

To examine how interactions from the current state-of-the-art CCSB Y2H pipeline behave in the GPCA assay, we examined two recent CCSB studies: The first study (Sahni et al, *Cell* 2015) – which examined interactions for both mutant and wild-type ORF – a subset of Y2H pairs involving wild type ORFs pairs were validated at the rate of 97/165 (59%) for Y2H-positive pairs and 7/17 (41%) for y2H-negatives. The second study (Hill et al, *Genes Dev* 2014) reported that GPCA validated 35% of Y2H-positive hits and 0% of Y2H-negative pairs. Thus, interactions emerging from BFG-Y2H are validated at a rate that is at least on par with current Y2H using the GPCA assay in human cell lines. We have now included this analysis in the main text.

Reviewer Comment 2.7

The authors then scaled up to larger interaction matrices and compare these results to traditional Y2H, finding that the barcode-based method performed worse, in most cases identifying only ~50% of the interactions of traditional methods (with an expected lower false positive rate).

Response 2.7

Comparing the quality of our four datasets with the latest Y2H dataset from Rolland et al (*Cell* 2014), we found that our recall rate was indeed lower but that precision was higher. According to the Matthews Correlation Coefficient performance measure – which combines recall and precision – BFG-Y2H screens performed on par with Rolland et al dataset at all scales up to the largest ~1,500 x 1,500 protein pair screen (Figure 7).

Reviewer Comment 2.8

Overall, I found this manuscript to be a frustrating read that lacks or never clearly articulates the information that is necessary to make a critical judgment of the method. One addressable problem is that the manuscript is filled with an enormous number of acronyms, many of which are unnecessary, and many others of which might be appropriate in specialist journals, but not for a general audience. I often found myself hunting all over the manuscript to decipher a single sentence that contains 5 or 6 acronyms, or a figure with acronyms not described in the legend.

Response 2.8

We thank the reviewer for this comment and apologize for the lack of clarity. We have reduced the number of abbreviations and attempted more generally to make the revised manuscript more readable and accessible to a wider audience. We are happy to work with the Editor to make any further changes in style or presentation that might be needed.

Reviewer Comment 2.9

However, there appear to be much deeper problems, particularly in the experimental design. The authors plate their double plasmid pools at a low density and grow these for 2-3 days on selective plates, meaning any design with a high fitness will have a long time (an important but uncharacterized number of generations) to expand exponentially within the population. Thus, by the time the pool is sequenced, the vast majority of cells are a small sliver of the most fit strains. As a practical matter, these fittest strains will constitute the vast majority of sequencing reads, meaning that in order to detect the lower fitness interactions by sequencing, either an enormous amount of sequencing must be "wasted" on resequencing of the fittest strains, or a new pool must be constructed with these high fitness strains thrown out.

Response 2.9

BFG-Y2H is indeed a pooled-strain genetic selection system where faster growing cells are expected to represent a disproportionate share of the final population (indeed the method depends on increased representation of strains expressing the reporter gene).

And yet, the precision of the BFG-Y2H screens was higher than that of Rolland et al and the overall performance combining recall and precision was on par with Rolland et al (Figure 7), at a cost that is substantially lower than current Y2H methods (Appendix Figure S4).

Two factors of our design made this possible.

The first factor was an experimental design that sought to capture interactions across a wide range of growth fitness effects in the Y2H-selective media. In order to minimize the growth competition effect, we spread cells sparsely on solid plates and found that the competition effect in solid plates was much smaller than that by liquid selection, presumably because the larger colonies exhaust local nutrient supplies more rapidly, allowing the slower-growing colonies to ‘catch-up’. This served to reduce the abundance differential between high-fitness and moderate-fitness strains.

The second factor is the remarkably low per-read cost of next-generation sequencing, such that “wasted reads” have only a modest impact on the final overall cost.

We have clarified these points in the revised manuscript.

Reviewer Comment 2.10

Making matters worse, it appears as though the vast majority of the highest fitness strains are autoactivators, meaning that most of the sequencing costs will go to experimental noise. With no clear way to identify autoactivators a priori, it is unclear how this problem can be overcome, and it was undiscussed in the larger matrix screens. Even with autoactivators removed, this assay will always be biased towards characterizing the strongest protein interactions. This effect provides a potential explanation for its poor performance when compared to traditional Y2H and why retesting of 100 interactions resulted in only 50% retest rate (competition between only the fittest 100 strains will cause lowest fitness strains in this group to decline in frequency). Much of the rest of this manuscript is statistical methods built around this flawed experimental design in order to improve the predictive value. However, if the aim of this method is to expand throughput to characterize the full human interactome, this simply won't work because lower fitness interactions will always be missing.

Response 2.10

First, we note that the precision of interactions captured by BFG-Y2H was higher (even without filtering by retesting results) than that of Rolland et al (which did filter out retest-failed pairs) and the overall performance combining recall and precision was on par with Rolland *et al* even without removing auto-activators (Figure 7). In fact, BFG-Y2H enabled us to systematically estimate quantitative background levels and to identify protein interactions even for high-background baits (please see Response 2.2).

We next note that all protein interaction methods are more likely to detect strong interactions. Indeed, tandem affinity purification-mass spectrometry methods perform two washes, so that only the interactions with the lowest off-rates can survive.

Although K_d information for protein interactions is scarce and unstandardized with respect to critically important conditions (salt, pH etc), we attempted to assess the bias of BFG-Y2H towards strong interactions. We extracted from the PDB-CN database all available K_d values for the pairs that we tested. Only 21 K_d values were available. We detected interactions for 4 of these 21 pairs, having published K_d values of 13, 20, 40.8 and 1500 nM (see plot below). The distribution of the K_d s of the pairs that we detected was not significantly different from the distribution of K_d s for other pairs with a published K_d . Given the scarcity of the data we cannot draw any strong conclusions, and have therefore not included this analysis in the paper. However, we describe it here in order to show that we did make our best effort to respond on this point.

Finally, we note that our pairwise Y2H retest success rates of ~50% were in accordance to that observed with the Harvard/Dana-Farber CCSB Y2H pipeline (Rual et al *Nature* 2005; Rolland et al, Cell 2014). Our validation rate by the orthogonal GPCA assay was 74% for the CENT screen and 46% for the CCC screen, which is also consistent with (if not better than) validation rates of interactions from the CCSB pipeline (see Response 2.6). We have also clarified the impact of heterogeneous growth effect on the screening procedure in the main text (please also see Response 2.9).

Additional major criticisms

Reviewer Comment 2.11

1) Given that the ultimate readout of this assay is sequencing count for each fusion construct, there is surprisingly little discussion of the methods and caveats around deep sequencing of fused plasmids. Missing is the PCR methods (cycles, etc.), number of reads for each experiment, and the analysis pipeline.

Response 2.11

We appreciate this comment and have now further clarified the methods, including the PCR protocol. The number of reads for each experiment is included in Appendix Data S3 and the analysis pipeline is described in Appendix Note S4. We also now explicitly state that all scripts used for data analysis are available upon request.

Reviewer Comment 2.12

How much of a problem is PCR jackpotting? Are sequencing errors ignored or clustered to a known barcode? How much sequencing error is tolerated? Does count error derived from sequencing resemble Poisson sampling? Does the size of the fusion impact what gets PCR amplified? These questions speak directly to the scalability of the system.

Response 2.12

We have designed the scoring system so that the impact of jackpotting effects and noise can be reduced by requiring agreement from replicates for each protein pair (Appendix Note S4). The sequencing reads were mapped to the pre-identified barcodes with an E-value threshold of $1e-10$. The barcode sizes are the same and potential PCR bias was normalized using non-selective (+His condition) abundance data. We clarified these points in the manuscript.

All theoretical concerns aside, the true practical test of scalability is an independent measure of assay performance at scale. Our study shows clearly that the method is performing well at the largest 1500 x 1500 scale. Performance at this scale means that the method is applicable at genome scale. Even with a fairly standard lab environment we already have the capacity of perform 15-20 BFG screens in parallel, so that it not difficult to image 100-200 repeats of a 1500x1500 screens space to achieve genome scale.

Reviewer Comment 2.13

2) Figure 3a shows a uniform distribution of fused barcode copy numbers. This is a crucial distribution that speaks to how many interactions one would never have a chance to see at various library sizes. One would naively expect that this should be roughly normal in His⁺, as each fusion barcode should have an ~ equal chance of making it in the pool without any selection. A uniform distribution is unexpected and never explained. Why is 0 excluded from these plots? Zero is the most important data point to interpreting how well the system works.

Response 2.13 As we noted regretfully in Response 2.1, the reader may have been lead to think that the distributions in Figure 3 (now Figure 4) were uniform, but this was not the case. Relative abundance of fused-barcode counts was previously shown with a log-scaled Y-axis, which de-emphasizes peaks in a distribution. We now show these plots with a linear Y-axis, which makes clear that the counts distribution in the non-selective condition was non-uniform.

According to our Monte-Carlo simulation, the distributions of +His and -His conditions were predicted to behave like this if strain abundances in the initial haploid pools follow a log-normal distribution with CV=30% due to clone-specific fitness effects in the pre-culture step (Appendix Note S3 and Figure S1).

Reviewer Comment 2.14

3) *There needs to be a better characterization of how all of the steps of making the double plasmid libraries alters barcode frequencies. How many generations in each step? How does noise introduced during this process impact the screen?*

Response 2.14

We completely agree that further optimization could permit further gains. However, we respectfully suggest that the optimization studies requested would unreasonably delay the first report of a new technology that has already been shown (through demonstrated high overall performance of two versions of the BFG-Y2H assay applied to four test spaces) to substantially outperform current methods in terms of throughput and cost, with similar quality.

Reviewer Comment 2.15

4) *The authors need to address more quantitatively the limitations of their screen. Crucially, how many generations pass during competitive growth in -His? What is the fitness difference on -His between a strain with a strong interaction vs. one with no interaction. Based on these estimates, in a screen with only these two strains, what fraction of the population would one expect to be the strong interaction strain at the time of sequencing? Can simulations be used to understand which lower-fitness interactions would not be detected?*

Response 2.15

We have estimated noise introduced throughout the screening steps by Monte-Carlo simulation which supported the rationale of the experimental design and its results (Appendix Note S3 and Figure S1). We do expect that the simulation framework could be used to further optimize the method, but would again respectfully suggest that extensive optimization studies would delay reporting of a method with substantial demonstrated advantages over current methods.

Reviewer Comment 2.16

5) *Figures and figure legends are often incomplete.*

Response 2.16

We apologize for this, and have now double-checked and clarified the figures and figure legends.

Reviewer Comment 2.17

6) *I would expect that plasmid size has an impact on the large matrix assays. For example large plasmids might be underrepresented in cell pools because it is more difficult to insert them into cells. Plasmid copy number variation may also be in issue. This could mean that many PPIs have no chance of being detected. However, these caveats are never discussed or addressed.*

Response 2.17

We appreciate the suggestion. Although Figure 6d shows that 90% of the pooled ORF clones were detected in the BFG-Y2H screening and the attrition during the yeast transformation was not high, we appreciate the reviewer's concern. We related ORF length to the probability that an ORF in the original search space was ultimately present in the non-selected BFG-Y2H pool and did not find a significant correlation during the yeast transformation. We observed a slight bias in ORF length during the *en masse* Gateway reaction, suggesting an iterative *en masse* Gateway strategy to cover a given space. We now include this information in Figure 6d and it is discussed in the main text.

Reviewer Comment 2.18

7) *"Accurately estimating the effects of Y2H selection on each X-Y pair must account for several factors: 1) uneven strain abundance in the initial DB-X and AD-Y libraries; 2) potential growth effects of DB-X and AD-Y expression; and 3) slight barcode-dependent differences in amplification and sequencing." In addition random growth processes, PCR jackpotting, and sampling from the sequencer are important and should be discussed.*

Response 2.18

Our procedure already accounts for systematic biases like uneven strain abundance, growth effects, amplification and sequencing bias of barcodes. This accounting is handled by rescaling the read counts observed post-selection by estimated read counts from the non-selective (+His) condition, which should reflect the same biases. Products of random errors such as jackpotting effects and low

counts were mitigated by use of internal replicates for each protein pair (Appendix Note S4). We have now clarified these points in the manuscript.

Minor criticisms

Reviewer Comment 2.19

IS stands for "interaction score", yet IS score (interaction score score) is used throughout the paper and figures

Response 2.19

Thank you. This has been corrected.

Reviewer Comment 2.20

Page 10 - "it might be applied to discover genetic interactions using high-multiplicity-of-infection lentiviral transduction to express or delete combinations of genes." Untrue in its current form. Genetic interactions require a fitness measurement, but this assay has a digital readout of whether each strain is in the pool or not.

Response 2.20

The quantitative interaction scores that BFG reports are based on quantitative estimates of the frequency of strains based on fused barcode tag abundance in two populations – one population that is under growth selection in the absence in supplemented histidine, and a reference population that is under growth selection in the presence of supplemented histidine. It is well established that barcode sequencing can provide estimates of relative fitness, and this has been applied many times since its first report (Smith et al, *Genome Research* 2009; with authors that overlap the present study). The present study extends the ideas of Smith *et al.* from barcode analysis of a single strain library to fused barcode analysis for a combinatorial library. Although relative fitness measurements are used here to detect protein interactions, we can see no conceptual barrier preventing the application of this technique to genetic interactions.

Reviewer Comment 2.21

Figure 3a - 10^{-2} appears twice on the x axis

Response 2.21

Thank you. Now fixed.

Reviewer Comment 2.22

Figure 3b, c - The blue color scale makes it extremely difficult to identify positives, or see differences

Response 2.22

We agree that it is difficult to pick out individual positives, but our purpose here was to represent the results schematically rather than as a means of conveying each individual result. Indeed, the individual scores for particular protein pairs are provided in supplementary tables. To this, we have now added information on raw barcode counts in the selective condition and estimated barcode counts in the non-selective condition (Appendix Data S3). For a schematic graphical representation, we tried many different color scales and felt that heat map representations with the blue color scale represented the data most effectively. However, we will consult with the Editor and would be happy to remove or alter the figures at their discretion.

Reviewer Comment 2.23

Figure 3d - axis labels of the number of reads are critical to interpreting these plots

Response 2.23

We apologize for the omission. We have added axis labels and clarified in the figure legend that the units are log scale measurements of barcode count divided by the total number of barcode counts in the relevant population.

Reviewer Comment 2.24

Figure 3f - insufficient legend

Response 2.24

Apologies. We have added a more complete legend.

Reviewer Comment 2.25

Figure 5c - A histogram is more straightforward and interpretable than a cumulative % plot

Response 2.25

The main purposes of this plot were to demonstrate which fraction of ORFs achieved at least one barcode (allowing its measurement by BFG-Y2H) and what fraction achieved at least two barcodes (allowing internal replication). The cumulative representation achieves this, which also showing the fraction of ORFs for which internal replication in triplicate, quadruplicate etc is possible. We have clarified this point in the figure legend.

Reviewer Comment 2.26

Figure EV2b - RY and Y or unexplained

Response 2.26

We have clarified the genotypes, origins and uses of RY and Y-strains in the figure legend.

Reviewer Comment 2.27

Figure EV2d - I'm confused. Are strains each containing a bait and a prey mated?

Response 2.27

Yes, we had a typographical error and said 'Mating' where we meant 'Pooling'. We have corrected this.

Reviewer Comment 2.28

Appendix S4e- Sequencing costs of the library are missing

Response 2.28

We have added the sequencing costs.

Reviewer #3:

The manuscript, entitled "Pooled-matrix protein interaction screens using Barcode Fusion Genetics" by Yachie et al, describe a new yeast two-hybrid method that enables multiplexed, one pot screen for protein-protein interactions. In their Barcode Fusion Genetics-Yeast Two-Hybrid (BFG-Y2H), the bait library (i.e., DB-X) is prepared in a haploid yeast strain in which each strain carries a plasmid that expresses a DB-X protein upon doxy induction and also contains UPTAG DNA barcode flanked by direction LoxP and Lox2272 recombination sites. Similarly, the prey library (i.e., AD-Y) is prepared in a haploid yeast strain in which each strain carries a plasmid that expresses an AD-Y protein upon doxy induction and also contains DNTAG DNA barcode flanked by direction LoxP and Lox2272 recombination sites. To screen for protein-protein interactions, the strains of these two libraries are pooled and mated en mass to create diploid cells, representing all X-Y combinations between the bait and prey proteins. When a pair of protein interacts in the diploid cells, the strain becomes His⁺ and survives on the His dropout media. By turning on Cre, recombination happens among LoxP and Lox2272 sites, generating an "Up-Up" barcode on the AD-Y plasmid and a "Dn-Dn" barcode on the DB-X plasmid. Using deep-seq, the authors could readily identify the pairs of positive PPI relationships. To demonstrate its application, the authors first tested a set of human centrosomal and centrosome-associated proteins, comprised of 143 DB-X and 162 AD-Y. Using non-selective media, they estimated that the screen was close to saturation and the sampling sensitivity was high. To score protein-protein interactions on the selective media, they normalized the sequencing read counts to obtain a quantitative interaction score (IS). They concluded that protein pairs with high IS scores are enriched for known interactions. Using another set of protein constructs implicated in cancer, the authors evaluated the performance of this BFG-Y2H approach in detecting auto-activators, those that do not require the prey to be able to activate His gene. They concluded that this new approach was sensitive enough to detect homodimer formation. They then went on to demonstrate a more efficient en mass strategy for producing barcoded BFG-Y2H constructs and compared it with the traditional Y2H approach.

In my opinion, this new design of pooled Y2H screen is ingenious because it allows simultaneous screens for any combinations of potential protein-protein interactions. In the past ten years, Vidal and colleagues have improved the state-of-art Y2H approach and applied it to map out global protein-protein interactions that laid the foundation of biological networks for many research fields. However, it is still quite labor-intensive and time-consuming. For this very reason, the human protein-protein interaction networks are still far from completion. This new BFG-Y2H approach now holds the potential to survey the entire 20,000 by 20,000 combinations of protein-protein interactions in humans, as well as in other important model organism. I envision it will be a tremendous boost to the entire biological community. I was also quite impressed by their careful design and test of this new approach. The comparison with the state-of-art Y2H approach serves as a solid validation for this new methodology.

Reviewer Comment 3.1

One point the author raised in the manuscript was very intriguing to me; that is the IS scores could indicate the affinity of a protein-protein interaction as it represents the normalized deep-sequencing read counts. I wonder whether the authors have taken protein pairs of either high versus low IS scores and measure their true binding affinity using standard methods, such as SPR and OCTET? In my opinion, if it holds true, it would be very valuable information to have for any pairs of protein-protein interactions. Other than this point, I don't have any more issues. This work suits very well with the readership of MSB.

Response 3.1

We greatly appreciate the positive feedback on our work and the constructive comments. It is a great idea to systematically measure the affinity of interaction pairs to assess if they correlate with our interaction scores. However, we do not expect that this relationship will be generally straightforward, given that the Gal4 DB and AD tags will sterically and allosterically interfere with the interaction to varying degrees. Thus, to properly explore this question would require the determination of a K distribution for each class of interaction scores. This in turn would require extensive protein purification and many months of characterization, delaying the release of the BFG-Y2H method to the community. We therefore feel that, although this is an extremely interesting question, it falls beyond the scope of our paper. We hope that the modeling of our detected protein interfaces and indication of the increased residue contacts in higher interaction scores will serve as at least suggestive evidence of a relationship between interaction strength and our interaction score (Figure 5f).

We did also attempt to examine interaction scores for K_d values mined from the literature (see response 2.10 above). However, the general scarcity of K_d data made it difficult to draw firm conclusions. We will continue to explore this direction, but respectfully suggest that the difficulty and time-consuming nature of affinity measurement experiments makes this line of investigation worthy of a separate study.

2nd Editorial Decision

09 March 2016

Thank you again for submitting your work to Molecular Systems Biology. We have now heard back from the three referees who were asked to evaluate your manuscript. As you will see below, referees #1 and #3 are satisfied with the modifications made and think that the study is now suitable for publication. However, reviewer #2 still raises a number of issues, which should be addressed in a revision of the manuscript. We do not think that further experimental analyses are required at this stage, but we would like to ask you to provide some further clarifications regarding the remaining issues in a revision of the manuscript and a point-by-point response.

REFeree COMMENTS

Reviewer #1:

The manuscript has greatly improved. One final comment: the yeast one-hybrid papers (e.g., PMID: 23791784) also include Y2H assays - pairwise. These can more easily detect weaker interactions that are likely missed by Y2H assays that use pooling. I think it is important to mention this in the comparisons. Aside from that I am happy with the revision.

Reviewer #2:

The authors have addressed my concerns regarding readability of the manuscript. It is greatly improved. However, the authors failed to sufficiently address many major criticisms that were brought up in the last round of review.

--I am quite confused by the following statement in the methods (and a similar statement in the response to reviewers): "Solid agar plates instead of liquid media was used for selection to reduce the growth competition among His⁺ strains: On an agar plate, slower-growing colonies are allowed to 'catch up' as colonies from faster-growing strains exhaust their local nutrient supplies." pg 17 First, I find no evidence in the manuscript that it is true that competition is reduced on agar during their protocol. This is precisely the sort of evidence that I requested in the last round of reviews -- experiments that measure the extent of competition between strains (particularly with autoactivators) and how this may limit the sensitivity of their assay. Reviewer 3 appeared to agree: "In my opinion, as an important control for the authors to address the fitness concern, they should take at least three pairs of proteins of known high, medium, and low affinity values and test their growth rate after mating in a pool." As far as I can tell, no experiments were performed along these lines to address our concerns.

Second, the proposition that slower-growing colonies 'catch-up' with faster-growing colonies on agar appears to contradict results of traditional YTH, protein fragment complementation, and SGA, which rely precisely on these differences in colony sizes to quantify effects. The very fact that a few strains dominate read counts (whether or not autoactivators are included in the pool) suggest that competition between strains is intense and that there are huge differences between the number of cells in each colony.

Third, I'm having trouble understanding how their reported protocol did indeed result in colonies being "spread ... sparsely on solid plates" (response 2.9). In the methods and Note 3, the authors report spreading $\sim 1e8$ diploid cells ($1e10$ total cells) per 150mm plate, with >10 copies of each X-Y pair plated. The apparent number of colonies per plate is not reported. However, between ~ 400 and ~ 600 hits are discovered in the CV and CVA screens (Figure 7d, and one might expect this to be an underestimate if competition is strong) with for 4 BC-BC versions of each hit being in the pool. Given these numbers, between 16000 ($400 \times 10 \times 4$) and 36000 colonies are expected per plate (confluent) even when ignoring false positive colony growth. Additionally, the authors state that 5-10% of baits are auto-activators (Response 2.2) suggesting that the false positive rate would be on the order of $1e5$ colonies per plate. Even if the number of observed number of colonies is far fewer than these estimates, this might be precisely because competition is keeping many colonies so small that they go unobserved.

Fourth, the authors argue that the "proof is in the pudding" because they identify a number of PPIs in large screens and that characterizing the impact of competition would constitute an undue burden on publication (Responses 2.14, 2.15). I disagree on both counts. First, the authors show the the pooled protocol identifying only $\sim 50\%$ of the interactions of traditional methods (with an expected higher precision). The most likely explanation is competition and auto-activators causes many false negatives. Second, no reviewer has suggested that authors further optimize the protocol and repeat the screens using these new methods (which, I agree would cause unnecessary delays). Rather we ask that the authors examine the important caveats of the screens that have already been performed, and the characterize the limitations of this assay. These experiments are quite straightforward and can be done with a few strains (e.g. reviewer 3's suggestion). As one example, characterizing the number of generations in each step (comment 2.14) is fairly trivial and can be done through cell counting on a hemocytometer and colony counting on plates and various stages of the procedure. I find it odd that the authors would be reluctant to perform these experiments, given that might both explain the high false negative rate and point a way forward to improving their assay in future screens. Competition has a direct impact on the reproducibility and scalability of their assay and should not go unaddressed.

--Figures 4a and EV4d. The authors never sufficiently explain what "low saturation" and "inferred" mean. I assume "low saturation" refers to a low expected coverage per barcode pair and that "inferred" means counting each half of a barcode pair in the pool, and then inferring what should be present if one assumes perfect mating. However, there is no reason to assume perfect mating. Indeed, the "low saturation" distribution varies too widely from the inferred for this to be true. Thus, the "inferred" distribution is misleading and should be removed.

--I made the point earlier that the vast majority of sequencing reads are wasted due to auto-activators, but it is unclear how much. As part of their response, the authors claim they "might" save \$1000 if all auto activators are removed and include new figures EV4a,b as proof. However, the bins of these figures obscures the answer I am seeking because some bins are too narrow, too broad (e.g. $>1e3$), or missing. I appreciate that the authors now included the read count data as supplemental material, which makes it feasible to get the answer (although one has to assume how many reads go to unswapped BC-BC pairs). But, without doing the analysis myself, I would like to know what is the distribution of reads across all barcode pairs.

-- Response 2.11 remains insufficient. How much template is used for the PCR? This is a crucial number to determine how many potential copies of each BC-BC pair could be amplified.

-- Response 2.12 -- What does an E-value threshold of $1e-10$ mean? Approximately how many mismatches are tolerated? I find no discussion of the potential impact of PCR jackpotting anywhere in the manuscript.

-- Response 2.18 -- Growth effects are unlikely to be comparable between the His⁺ and His⁻ conditions. "Products of random errors such as jackpotting effects and low counts were mitigated by use of internal replicates for each protein pair" -- I have trouble understanding if this is true given that the author's scoring statistic uses only the top 5 scores to calculate and interaction. That is, the vast majority of replicates are ignored.

-- R values in Figure 4d and the text contradict each other.

-- Response 2.20. Comparative bar-seq (e.g. Smith 2009) results in a rank for each barcode but not a fitness (defined as the change in per generation fitness compared to the unperturbed cell). The rank will change depending on the pool of genotypes that a barcode is competing against, but the fitness will not. Accurate genetic interaction screens require a fitness for each single and double KO, not a rank. This manuscript does not show that this protocol can be used to measure fitness, so this claim overstates what is possible.

Reviewer #3:

One of my major concern was a potential correlation between the high sequence reads of a detected PPI and the affinity value of their interactions. I suggested that the authors select one pair from the high, medium, and low PPI categories and test their affinity value. Though the authors argued that it would be beyond the scope of their study, they did search the literature and identified a dozen reported affinity values of their identified PPI pairs. By comparing to their dataset, they did not find any statistically significant correlation. Although it is not a direct approach, I am happy to see this new result. After reading their rebuttals to Reviewers 1 and 2, I think the authors have addressed most of the previous concerns rather well.

2nd Revision - authors' response

11 March 2016

We thank the editor and reviewers for their constructive comments. We hope that the point-by-point responses below satisfy the remaining reviewer concerns.

Reviewer #1:*Comment 1.1*

The manuscript has greatly improved. One final comment: the yeast one-hybrid papers (e.g., PMID: 23791784) also include Y2H assays - pairwise. These can more easily detect weaker interactions that are likely missed by Y2H assays that use pooling. I think it is important to mention this in the comparisons. Aside from that I am happy with the revision.

Response 1.1

Thank you for this suggestion. The latest revision now includes the pairwise Y2H methods described in the yeast one-hybrid papers in the method comparisons table (Table EV4).

Reviewer #2:*Comment 2.1*

The authors have addressed my concerns regarding readability of the manuscript. It is greatly improved.

Response 2.1

Thank you.

Comment 2.2

However, the authors failed to sufficiently address many major criticisms that were brought up in the last round of review.

--I am quite confused by the following statement in the methods (and a similar statement in the response to reviewers): "Solid agar plates instead of liquid media was used for selection to reduce the growth competition among His⁺ strains: On an agar plate, slower-growing colonies are allowed to 'catch up' as colonies from faster-growing strains exhaust their local nutrient supplies." pg 17

First, I find no evidence in the manuscript that it is true that competition is reduced on agar during their protocol. This is precisely the sort of evidence that I requested in the last round of reviews -- experiments that measure the extent of competition between strains (particularly with autoactivators) and how this may limit the sensitivity of their assay. Reviewer 3 appeared to agree: "In my opinion, as an important control for the authors to address the fitness concern, they should take at least three pairs of proteins of known high, medium, and low affinity values and test their growth rate after mating in a pool." As far as I can tell, no experiments were performed along these lines to address our concerns.

Response 2.2

We apologize for not being more clear on this point. Of course we expect that cells expressing more of the *HIS3* reporter gene will grow faster, whether this is due to a higher concentration of the interaction-dependent reconstituted Gal4 complex, or due to an ability of the bait fusion to recruit RNA Pol2 directly ('auto-activation'). In fact, every pooled Y2H assay, in order to select for strains corresponding to interactions, absolutely requires that cells with increased *HIS3* expression grow faster.

Although we can offer a rationale for our choice of performing pooled growth on solid media, we did not perform experiments to directly assess the impact of this choice on performance. Here we attempt to describe that rationale more clearly:

The rationale for performing growth competitions on solid instead of liquid media stems from our desire to reduce the abundance advantage of the fastest-growing cells. In liquid media, we expect that all strains, after lag phase, will initially grow exponentially. Although this growth is dampened over time by reduced concentrations of nutrients and increased concentrations of catabolites in the media, the time course of these growth-dampening media changes should be common to all cells in

well-mixed liquid culture. On solid media, we expected that growth of the fastest growing cells tends to saturate slightly earlier than it would for slower growing cells. The intuitive explanation for this is that faster-growing colonies should tend to ‘contact’ neighboring colonies slightly earlier than would slower-growing colonies. (In this context, we consider “contact” to include proximities that are close enough for colonies to rob one another of diffusible nutrients.)

Moreover, models of colony growth suggest that growth is exponential at the periphery and sub-exponential at the interior of a colony, so that (after the earliest divisions) the overall growth of cells on solid media over time is sub-exponential (Jönsson and Levchenko, *Multiscale Model. Simul.* 2015)

Although we could continue debating our rationale for choosing solid vs liquid media, we have now removed all mention of the relative merits of solid vs liquid media from the paper. This question of whether and why to use solid vs liquid media seems like a distraction from the main point of the paper—description of a new technology that is demonstrated to efficiently identify protein interactions at a cost which is substantially decreased relative to state-of-the-art Y2H methods.

Comment 2.3

Second, the proposition that slower-growing colonies 'catch-up' with faster-growing colonies on agar appears to contradict results of traditional YTH, protein fragment complementation, and SGA, which rely precisely on these differences in colony sizes to quantify effects. The very fact that a few strains dominate read counts (whether or not autoactivators are included in the pool) suggest that competition between strains is intense and that there are huge differences between the number of cells in each colony.

Response 2.3

We hope that the rationale given in Response 2.2 has now made clear that we completely agree with the reviewer that there are huge differences in abundance between different strains. Our rationale for using solid media was to compress these huge differences to some degree.

We also agree that read counts are dominated by the most abundant strains, but want to make clear that sequencing is economical enough that, despite this phenomenon, we can still use sequencing to measure the abundance of strains across a wide range of growth rates.

RNA-seq technology offers a useful analogy here: The read counts in an RNA-seq experiment are dominated by the most abundant genes, and yet it is still possible to measure expression levels across a wide dynamic range of expression levels. Of course one could complain that RNA-seq is ‘wasting reads’ by devoting the lion’s share of sequencing resources towards counting the most abundant genes over and over again. However, this complaint would not change the fact that RNA-seq is already an extremely useful and economical technology, owing to the low per-read cost of sequencing.

Comment 2.4

Third, I'm having trouble understanding how their reported protocol did indeed result in colonies being "spread ... sparsely on solid plates" (response 2.9). In the methods and Note 3, the authors report spreading $\sim 1e8$ diploid cells ($1e10$ total cells) per 150mm plate, with >10 copies of each X-Y pair plated. The apparent number of colonies per plate is not reported. However, between ~ 400 and ~ 600 hits are discovered in the CV and CVA screens (Figure 7d, and one might expect this to be an underestimate if competition is strong) with for 4 BC-BC versions of each hit being in the pool. Given these numbers, between 16000 ($400 \times 10 \times 4$) and 36000 colonies are expected per plate (confluent) even when ignoring false positive colony growth.

Response 2.4

Please note that we used 20 plates (each 150mm in diameter) for CV and CVA screens (this point has now been clarified in the manuscript), so that the reviewer’s estimation of colony number could be reduced roughly 20-fold. However, the reviewer is correct that, at the time of harvesting, colonies were relatively confluent. We hope that the rationale given in Response 2.2 now makes clear that our point was that the faster-growing colonies will tend to contact neighboring colonies earlier than will slower-growing colonies. More importantly, we have removed all statements and claims relating to the choice of solid vs. liquid media.

Comment 2.5

Additionally, the authors state that 5-10% of baits are auto-activators (Response 2.2) suggesting that the false positive rate would be on the order of $1e5$ colonies per plate. Even if the number of observed number of colonies is far fewer than these estimates, this might be precisely because competition is keeping many colonies so small that they go unobserved.

Response 2.5

We agree that, just as there are genes with abundance too low to be detected by an RNA-seq experiment at a given sequencing depth, there will also be strain growth levels that are low enough to preclude observation. However, for the strains that are represented well enough on the plate to be quantified by sequencing, we can estimate the ratio of relative abundance of each strain before and after the selection. It is true that high-background baits ('autoactivators') will show increased relative abundance in combination with all preys, but these do not automatically lead to false positive interactions owing to our subsequent step of separately rescaling scores for each bait. When we carried out the CENT screen with and without 'autoactivators', we achieved a similar performance. This supports the idea that high-background baits do not yield false positive interactors at an appreciably higher rate than do other baits.

Comment 2.6

Fourth, the authors argue that the "proof is in the pudding" because they identify a number of PPIs in large screens and that characterizing the impact of competition would constitute an undue burden on publication (Responses 2.14, 2.15). I disagree on both counts. First, the authors show the the pooled protocol identifying only ~50% of the interactions of traditional methods (with an expected higher precision). The most likely explanation is competition and auto-activators causes many false negatives.

Response 2.6

It is true that the recall we achieved here with BFG-Y2H was lower than that of current Y2H methods, achieving a relative recall of ~50% for some screens (but a relative recall of 83% for the CV screen). The reviewer suggests that higher precision is an automatic consequence of lower recall, but this is not necessarily the case. For example, in current Y2H methods reducing the number of replicate screens achieves lower recall at the same precision. Here, for the two EMLR screens, we nearly double the precision of current Y2H methods. The overall performance as shown by the Matthew Correlation Coefficient (which balances both recall and precision) shows that we perform en par with the state-of-the-art.

Comment 2.7

Second, no reviewer has suggested that authors further optimize the protocol and repeat the screens using these new methods (which, I agree would cause unnecessary delays). Rather we ask that the authors examine the important caveats of the screens that have already been performed, and the characterize the limitations of this assay. These experiments are quite straightforward and can be done with a few strains (e.g. reviewer 3's suggestion). As one example, characterizing the number of generations in each step (comment 2.14) is fairly trivial and can be done through cell counting on a hemocytometer and colony counting on plates and various stages of the procedure. I find it odd that the authors would be reluctant to perform these experiments, given that might both explain the high false negative rate and point a way forward to improving their assay in future screens. Competition has a direct impact on the reproducibility and scalability of their assay and should not go unaddressed.

Response 2.7

We appreciate that the reviewer is not asking for any new screens to be performed.

We also completely agree that the proposed experiments represent one of many promising directions to further characterize the BFG-Y2H method that could potentially lead to increased future performance.

However, we have already shown that our method performs well, and is reproducible and scalable to at least a matrix of 2.6 million pairs, at a substantial cost reduction compared to current methods.

We respectfully suggest that further optimization of this and other parameters of the BFG-Y2H method not only warrants extensive future study, but also a future publication. With regard to measuring the number of generations at each step, we now clarify the OD₆₀₀ threshold values that we used as guides on when to proceed to the next step (Materials and Methods).

Comment 2.8

--Figures 4a and EV4d. The authors never sufficiently explain what "low saturation" and "inferred" mean. I assume "low saturation" refers to a low expected coverage per barcode pair and that "inferred" means counting each half of a barcode pair in the pool, and then inferring what should be present if one assumes perfect mating. However, there is no reason to assume perfect mating. Indeed, the "low saturation" distribution varies too widely from the inferred for this to be true. Thus, the "inferred" distribution is misleading and should be removed.

Response 2.8

To reduce sequencing costs low, we sequenced the +His pool at a depth sufficient to accurately measure the marginal abundance of the prey and bait barcodes. Although this depth is sufficient for measuring marginal abundance of bait and prey barcodes, we use the term 'low saturation' to make clear that this coverage is not sufficient to provide an accurate direct measure the abundance of fused barcode pairs represents the actual distribution observed from the sequencing of each barcode pair. As the reviewer suggests, we infer the abundance of the fused barcodes by assuming that cells of every strain are equally likely to mate with cells of any other strain, and thus use the term 'inferred' to refer to fused barcode counts estimated from the marginal abundance of bait and prey barcode counts. We have updated the latest revision to better explain these terms. We have added explanations for these terms in the figure legends.

It would be very interesting to investigate the accuracy of our assumption of uniform mating probabilities, and the impact on performance of any deviations from independence. However, given that we have already demonstrated performance of BFG-Y2H on par with current Y2H, we respectfully suggest that the suggested analysis falls outside the scope of the current study.

Comment 2.9

--I made the point earlier that the vast majority of sequencing reads are wasted due to auto-activators, but it is unclear how much. As part of their their response, the authors claim they "might" save \$1000 if all auto activators are removed and include new figures EV4a,b as proof. However, the bins of these figures obscures the answer I am seeking because some bins are too narrow, too broad (e.g. >1e3), or missing. I appreciate that the authors now included the read count data as supplemental material, which makes it feasible to get the answer (although one has to assume how many reads go to unswapped BC-BC pairs). But, without doing the analysis myself, I would like to know what is the distribution of reads across all barcode pairs.

Response 2.9

We are confused by the reviewer's comment. Figure 4a and EV5d contain the distribution of the fused barcode abundance of all pairs in all conditions and include the non-selective (+His) condition and the inferred read counts (as defined above). We should note that no 'un-swapped' barcode pairs are sequenced because the primers used to amplify the barcode regions and add the sequencing tags are specific to the chimeric fused barcodes.

Comment 2.10

-- Response 2.11 remains insufficient. How much template is used for the PCR? This is a crucial number to determine how many potential copies of each BC-BC pair could be amplified.

Response 2.10

For CV and CVA screens, we queried >20 ng of DNA extracted from yeast cells treated with doxycycline for each of screening and non-screening conditions. According to our Illumina Nextera sequencing analysis, we estimate that more than 5% of the yeast Miniprep product was 10-kb Y2H vectors (Figure EV6), suggesting that 20 ng of the yeast Miniprep product contained 90 million Y2H molecules (see also Note EV3). This number should be sufficient to estimate marginal

abundance of every barcode pair in the non-screening condition and to represent all the Y2H-positive barcode pairs in the screening condition. We have added this information in the Materials and Methods.

Comment 2.11

-- Response 2.12 -- What does an E-value threshold of 1e-10 mean? Approximately how many mismatches are tolerated? I find no discussion of the potential impact of PCR jackpotting anywhere in the manuscript.

Response 2.11

An E-value is defined to be the number of matches one would expect by chance. For E-values $\ll 1$, the E-value is approximately the same as the probability of getting a match by chance. Thus, an E-value of 1e-10 reflects a very low probability of a random match and therefore a high probability that the match corresponds to our specific barcode sequence.

We agree that PCR jackpotting is a potential source of random error in our abundance estimate. It would be interesting to investigate the impact of this potential source of error, and if necessary to reduce it (e.g., using emulsion PCR approaches that have been demonstrated to reduce jackpotting). However, given that we have already demonstrated performance of BFG-Y2H on par with current Y2H, we respectfully suggest that this falls outside the scope of the current study. We have added mention of reduced PCR jackpotting as a direction of potential future improvements.

Comment 2.12

-- Response 2.18 -- Growth effects are unlikely to be comparable between the His⁺ and His⁻ conditions. "Products of random errors such as jackpotting effects and low counts were mitigated by use of internal replicates for each protein pair" -- I have trouble understanding if this is true given that the author's scoring statistic uses only the top 5 scores to calculate and interaction. That is, the vast majority of replicates are ignored.

Response 2.12

We respectfully disagree that the vast majority of replicates are ignored. This is akin to saying that calculating a median of 1000 numbers ignores all but the 1 or 2 central values. Just as with median estimation, our final score depends on the full distribution of replicate scores.

Comment 2.13

-- R values in Figure 4d and the text contradict each other.

Response 2.13

Thank you. We corrected this in the text.

Comment 2.14

-- Response 2.20. Comparative bar-seq (e.g. Smith 2009) results in a rank for each barcode but not a fitness (defined as the change in per generation fitness compared to the unperturbed cell). The rank will change depending on the pool of genotypes that a barcode is competing against, but the fitness will not. Accurate genetic interaction screens require a fitness for each single and double KO, not a rank. This manuscript does not show that this protocol can be used to measure fitness, so this claim overstates what is possible.

Response 2.14

We agree that Smith *et al.* does not demonstrate measurement of absolute fitness. We might argue that this work does demonstrate measurement of relative abundance and of relative fitness, and that these are sufficient information for measuring genetic interaction. However, there seems little point in the argument because our manuscript never claims that we can measure fitness or genetic interaction. There is only one section in our latest revision that mentions genetic interactions: "Although the BFG technology was developed to screen protein interactions, it is applicable to other context-dependent phenotypic screens involving multiple reagents amenable to barcoding. For

example, it might be applied to discover genetic interactions using high multiplicity-of-infection lentiviral transduction to express or delete combinations of genes.”

We hope that the reviewer agrees that this is a clearly-labeled speculation about a possible future application.

Reviewer #3:

Comment 3.1

One of my major concern was a potential correlation between the high sequence reads of a detected PPI and the affinity value of their interactions. I suggested that the authors select one pair from the high, medium, and low PPI categories and test their affinity value. Though the authors argued that it would be beyond the scope of their study, they did search the literature and identified a dozen reported affinity values of their identified PPI pairs. By comparing to their dataset, they did not find any statistically significant correlation. Although it is not a direct approach, I am happy to see this new result. After reading their rebuttals to Reviewers 1 and 2, I think the authors have addressed most of the previous concerns rather well.

Response 3.1

Thank you.

Corresponding Author Name: Frederick P Roth and Nozomu Yachie

Manuscript Number: MSB-15-6660R